# FEDERATED LEARNING WITH DIFFERENTIAL PRIVACY FOR END-TO-END SPEECH RECOGNITION

## ABSTRACT

While federated learning (FL) has recently emerged as a promising approach to train machine learning models, it is limited to only preliminary explorations in the domain of automatic speech recognition (ASR). Moreover, FL does not inherently guarantee user privacy and requires the use of differential privacy (DP) for robust privacy guarantees. However, we are not aware of prior work on applying DP to FL for ASR. In this paper, we aim to bridge this research gap by formulating an ASR benchmark for FL with DP and establishing the first baselines. First, we extend the existing research on FL for ASR by exploring different aspects of recent *large end-to-end transformer models*: architecture design, seed models, data heterogeneity, domain shift, and impact of cohort size. With a *practical* number of central aggregations we are able to train **FL models** that are **nearly optimal** even with heterogeneous data, a seed model from another domain, or no pre-trained seed model. Second, we apply DP to FL for ASR, which is non-trivial since DP noise severely affects model training, especially for large transformer models, due to highly imbalanced gradients in the attention block. We counteract the adverse effect of DP noise by reviving per-layer clipping and explaining why its effect is more apparent in our case than in the prior work. Remarkably, we achieve user-level $(7.2, 10^{-9})$-**DP** (resp. $(4.5, 10^{-9})$-**DP**) with a 1.3% (resp. 4.6%) absolute drop in the word error rate for extrapolation to high (resp. low) population scale for **FL with DP in ASR**.

## 1 INTRODUCTION

Federated learning (FL) allows training models in a distributed manner without storing data centrally on a server (Konečný et al., 2015). While FL on its own provides only limited privacy guarantees (Boenisch et al., 2023; Carlini et al., 2023; Kariyappa et al., 2023; Bertran et al., 2019; Azam et al., 2022), it can be combined with differential privacy (DP) (Dwork et al., 2014) and secure aggregation (Bonawitz et al., 2016; Talwar et al., 2023) to provide strong privacy guarantees for users (or clients) while training high quality models (Abadi et al., 2016). FL introduces a lot of challenges into the model training, e.g. heterogeneous data (Li et al., 2020; Wang et al., 2020), scaling laws for large cohort training (Charles et al., 2021), and convergence rate due to local training (Malinovsky et al., 2022). Moreover, for *practical* FL with DP we are limited by the total privacy budget that we can spend on hyper-parameter tuning, because it incurs additional overhead in terms of the privacy, and communication and computation cost (Wang et al., 2018; Azam et al., 2021). Thus, robust models and simple training recipes are of great interest.

Applying FL to train end-to-end (E2E) automatic speech recognition (ASR) models is also challenging (Guliani et al., 2021; Yu et al., 2021; Guliani et al., 2022; Gao et al., 2022; Nguyen et al., 2023) especially due to the inherently heterogeneous data (Cui et al., 2021; Gao et al., 2022) (uniform data sampling used for central training is impossible for FL) and models based on transformer architecture (Synnaeve et al., 2020; Baevski et al., 2020; Gulati et al., 2020; Kim et al., 2022). It is well known that to train high quality transformer models, we typically need to apply a lot of optimization tricks such as learning rate warm-up and decay, gradient clipping, adaptive optimizers, special initialization and others (Zhang et al., 2022b; Dehghani et al., 2023; Zhai et al., 2023). FL on its own provides limited privacy even in the context of ASR (Tomashenko et al., 2022; Nguyen et al., 2023). However, to the best of our knowledge, no prior work exists that incorporates DP into FL for

ASR. We argue that using DP in FL for ASR is hard but feasible, and provide the first benchmarks for FL with DP for ASR.

Most prior works on both FL and DP use small-scale models mainly due to (i) communication complexity of FL (Kairouz et al., 2021b) and (ii) difficulty of training large-scale models with DP (Bassily et al., 2014; Shen et al., 2021; Tramèr & Boneh, 2020). We argue that (i) practical model sizes are increasing over time, including ASR models (Botros et al., 2023), and (ii) optimization of larger (over-parametrized) models is simpler (Woodworth et al., 2020)[1]. To fill the gap in understanding larger scale models in the context of FL and DP, and to alleviate the optimization issues related to training smaller models, we solely study a large vanilla transformer model for ASR.

In this paper, we focus on (i) providing an empirical analysis of FL performance on E2E ASR using a large (250M parameters) vanilla encoder-based only transformer model trained with the Connectionist Temporal Classification (CTC) loss (Graves et al., 2006); (ii) analyzing the impact of key FL parameters, data heterogeneity, and seed models (pre-trained with or without domain shift) on FL performance in ASR; (iii) formulating a *practical benchmark* and establishing the *first baselines for FL with DP for ASR*; and (iv) reviving the per-layer clipping for efficient training of large transformer models even in the presence of DP noise. We show that FL can be used to train competitive, nearly optimal models compared to centrally trained models for several datasets, covering several languages (English, German, French): FL models are at worst $\sim 0.3\%$-$1.4\%$ absolute word error rate behind the corresponding central models with a limited number of central steps. Competitive models are obtained even with heterogeneous data, especially when the training starts from a seed model. The seed model can even come from another domain and perform relatively poorly on the target dataset. We also show that FL with user-level DP, which is more preferable to example-level DP, and large models is viable for E2E ASR and promising even for low-resource languages. With per-layer clipping, our models achieve $(7.2, 10^{-9})$-**DP** (resp. $(4.5, 10^{-9})$-**DP**) with $2.1\%$ (resp. $4.6\%$) degradation in absolute word error rate for extrapolations to high (resp. low) population scale for **FL with DP in ASR**.

## 2 RELATED WORKS

FL for ASR was first studied by Dimitriadis et al. (2020) who used attention-based sequence-to-sequence (Seq2Seq) LSTM models. The paper showed that FL performance in ASR suffers from the heterogeneous data distribution, which is a known problem in FL (Zhao et al., 2018; Kairouz et al., 2021b). The authors suggested a gradient weighting scheme to speed up convergence and improve the obtained models. Later, Cui et al. (2021) proposed to use FL with hybrid LSTM models and introduced client adaptive federated training to alleviate the impact of data heterogeneity by normalizing data across clients. Guliani et al. (2021) used RNN from Graves et al. (2013) and, to deal with data heterogeneity, proposed to add noise to gradients during the local training. However, the obtained FL models still performed poorly compared to their central training counterparts. In the concurrent work, Azam et al. (2023) showed that adaptive optimizers induce smoothness, which helps to overcome data heterogeneity. While our results confirm that data heterogeneity has a negative impact on the performance of FL for ASR, we argue that for transformer models trained with adaptive optimizers it does not prevent FL from finding competitive models even without introducing algorithms to specifically tackle data heterogeneity.

Moving to E2E ASR with transformers augmented with convolutions and $\sim$120M parameters (Gulati et al., 2020), Guliani et al. (2022) proposed to reduce communication complexity by introducing federated dropout (FD) and training only a subset of model parameters on each client. Besides reducing communication complexity, this also made FL models more similar to the central training counterparts. However, the training was typically run for tens to hundreds of thousands central steps, which is often impractical in real-world scenarios. In this paper, we restrict training to *practical 2k central steps* and show that FL can nearly match the central training. Furthermore, most of our experiments are done on Common Voice (CV) (Ardila et al., 2020) dataset, which is more heterogeneous than LibriSpeech (LS) (Panayotov et al., 2015) used in the aforementioned paper, and it allows us to study the performance and the robustness to hyper-parameters on multiple languages.

---

[1]For example, distillation from a large model into a small model is still the dominant method in the research community to train small models (Stanton et al., 2021).

In recent work, Gao et al. (2022) focused on a relatively small Seq2Seq model with CNN encoder and RNN decoder trained with joint CTC-attention objective using a sub-word tokenizer. First, the authors highlighted the necessity of switching from LS data to CV (French and Italian) as CV has a more realistic data distribution for FL. Second, they argued that it is "nearly impossible to train an E2E ASR model from scratch in a realistic FL setup" and proposed to use an additional training step over a small batch of held-out data on the server, after the FL model update. Like Gao et al. (2022), we focus on both LS and CV data. However, we show that FL training from scratch for *larger* models is viable with a proper optimizer and without the need for additional training steps on server data.

Jia et al. (2022) proposed an FL system for on-device ASR domain adaptation assuming user data are out-of-domain and without any ground-truth transcriptions (unlabeled data). Their work uses federated training starting from a pre-trained seed model. Nguyen et al. (2023) proposed to use a large (∼300M parameters) pre-trained self-supervised model (transformer) as the initial seed and significantly improved FL performance. The authors also found that model updates reveal information about the speakers. This provides support for the importance of using DP in combination with FL. In this paper, we assume that all central and user data are labeled and leave to future work the use of unlabeled data.

Audio data reveal information about the content but they can also be used to derive other pieces of sensitive information such as biometric identity, emotions, health condition and others (Kröger et al., 2020). A number of researchers have argued that FL does not really protect user privacy (Boenisch et al., 2023; Kariyappa et al., 2023), and several works (Tomashenko et al., 2022; Nguyen et al., 2023) discussed privacy attacks specifically for FL in ASR. However, to the best of our knowledge, there has not been any prior work on applying DP to FL for ASR, which is the main focus of this work.

## 3 FEDERATED LEARNING WITH DIFFERENTIAL PRIVACY

**Federated Learning (FL)**   In this paper, we focus on cross-device FL where only a small fraction $q = L/N$ of users (clients) participate in each step of central aggregation (*central step*), where $N$ is the total number of users (population) and $L$, termed *cohort size*, is the number of users uniformly sampled from all users during each central step. Users cannot maintain a state across rounds. Each user $n$ has its own local data $\mathcal{D}_n$ of paired audio and the corresponding ground-truth transcription. The objective of FL is to minimize the global loss function $\mathcal{L}(\mathcal{D}; \boldsymbol{\theta})$ given the ASR model parameters $\boldsymbol{\theta}$ and all user data $\mathcal{D} = \cup_{n=1}^N \mathcal{D}_n$: $\min_{\boldsymbol{\theta} \in \mathbb{R}^D} \left\{ \mathcal{L}(\mathcal{D}; \boldsymbol{\theta}) \triangleq \frac{1}{|\mathcal{D}|} \sum_{n=1}^N |\mathcal{D}_n| \mathcal{L}(\mathcal{D}_n; \boldsymbol{\theta}) \right\}$. FL optimizes $\mathcal{L}(\mathcal{D}; \boldsymbol{\theta})$ via sampling a set of users who perform local training on their own data $\mathcal{D}_n$ and periodically upload their model updates to the server, which aggregates the updates and performs the central model step either through conventional federated averaging (Konečný et al., 2016), or through an adaptive optimizer (Reddi et al., 2021). The updated central model is broadcasted to another sampled set of users and the process is repeated either for a fixed number of central steps $T$ or until convergence. The FL algorithm and the corresponding terminology are summarized in Algorithm 1.

**FL with Differential Privacy (DP)**   Since no prior work exists that applies private FL to ASR, we highlight its necessity and thus in the rest of the paper we *formulate a practical benchmark and establish the baselines for private FL in ASR*. To do so, we refer to DP (Dwork et al., 2006; Dwork, 2011; Dwork et al., 2014), which provides a mathematical formalism of guarantees on the amount of information learnt by machine learning models from the user private data:

**Definition 1** *Differential privacy: A randomized mechanism* $\mathcal{M} : \mathcal{D} \to \mathcal{R}$ *with a domain* $\mathcal{D}$ *(e.g., possible training datasets) and range* $\mathcal{R}$ *(e.g., all possible trained models) satisfies* $(\epsilon, \delta)$*-differential privacy if for any two adjacent datasets* $D, D' \in \mathcal{D}$ *and for any subset of outputs* $S \subseteq \mathcal{R}$ *it holds that* $Pr[\mathcal{M}(D) \in S] \le e^\epsilon Pr[\mathcal{M}(D') \in S] + \delta$.

One key component of the DP definition is **adjacent datasets**. In some applications, prior works consider the example-level privacy (Chaudhuri et al., 2011; Abadi et al., 2016). In the federated learning setting where each user has multiple data points, we would like to protect all of the data that a user contributes. Following McMahan et al. (2018), we therefore apply the definition of DP to protect user-level information by considering the following adjacency relation:

**Definition 2** *User-adjacent datasets: Let* $D$ *and* $D'$ *be two datasets of training examples, where each example is associated with a user. Then,* $D$ *and* $D'$ *are adjacent if* $D'$ *can be formed by adding or removing all of the examples associated with a single user from* $D$.

---

**Algorithm 1:** Federated learning with differential privacy (marked as red) for transformer model

---

**Inputs:** Initial model state $\boldsymbol{\theta}^0$ (either randomly initialized or pre-trained on server data), central steps $T$, central optimizer opt, clients sampling rate $q$, local steps $T_l$, local optimizer $\text{opt}_l$, clipping function $\text{clip}(\boldsymbol{v}, C) = \boldsymbol{v} \cdot \min\left(1, \frac{C}{||\boldsymbol{v}||}\right)$, local clipping $C_l$, DP clipping $C$ and DP noise $\sigma$.

**Result:** ASR model $\boldsymbol{\theta}^T$

1 Initialize central optimizer opt

2 **for** $t = \overline{1, T}$ **do**

3     Sample every client with probability $q$ to form a subset $\mathcal{N}^t$ of clients from all clients $\mathcal{N}$ ($|\mathcal{N}| = N$)

4     // For practical implementation we fix the size of the cohort $\mathcal{N}^t$ to $L$ throughout the training.

5     **for** $n = \overline{1, |\mathcal{N}^t|}$ **in parallel do**

6        Initialize local model $\boldsymbol{\theta}_n^{(t,0)} \leftarrow \boldsymbol{\theta}^{t-1}$ and local optimizer $\text{opt}_l$

7        **for** $t_l = \overline{1, T_l}$ **do**

8           // We also use local epochs instead of steps: then this loop has different number of steps per client.

9           Sample train mini-batch $\boldsymbol{B}_n^{t_l} \in \mathcal{D}_{\mathcal{N}_n^t}$ and compute gradient estimate $\boldsymbol{g}_n^{(t,t_l)}(\boldsymbol{B}_n^{t_l}; \boldsymbol{\theta}_n^{(t,t_l-1)})$

10           Clip gradients $\boldsymbol{g}_n^{(t,t_l)} \leftarrow \text{clip}(\boldsymbol{g}_n^{(t,t_l)}, C_l)$ and update a local model $\boldsymbol{\theta}_n^{(t,t_l)} \leftarrow \text{opt}_l(\boldsymbol{g}_n^{(t,t_l)})$

11        Compute client's delta $\boldsymbol{\Delta}_n^t = \boldsymbol{\theta}_n^{(t,T_l)} - \boldsymbol{\theta}_n^{(t,0)} = \boldsymbol{\theta}_n^{(t,T_l)} - \boldsymbol{\theta}^{t-1}$

12        Clip client's delta $\boldsymbol{\Delta}_n^t \leftarrow \text{clip}(\boldsymbol{\Delta}_n^t, C)$

13        Add Gaussian noise to client's delta $\boldsymbol{\Delta}_n^t \leftarrow \boldsymbol{\Delta}_n^t + \mathcal{N}(0, I\sigma^2 qN)$

14     Compute central model's pseudo-gradient $\boldsymbol{g}^t = \boldsymbol{\Delta}^t = \frac{1}{|\mathcal{N}^t|} \sum_{n=1}^{|\mathcal{N}^t|} \boldsymbol{\Delta}_n^t$

15     Update the central model $\boldsymbol{\theta}^t \leftarrow \text{opt}(\boldsymbol{g}^t)$

---

To incorporate user-level DP into FL, there are two additional steps shown in Algorithm 1: (i) clipping users' deltas $\boldsymbol{\Delta}_n^t$ so that they have bounded $L_2$ norm $\|\boldsymbol{\Delta}_n^t\|_2 \leq C$ at every central training step $t$; (ii) addition of Gaussian noise $\mathcal{N}(0, I\sigma^2 qN)$ to the clipped deltas before sending them to the server. We use the Gaussian moments accountant from Abadi et al. (2016) to achieve tight privacy bounds and restate the main theorem of McMahan et al. (2018) in our parametrization of Gaussian noise added to every user's model update before the averaging:

**Theorem 1** *For the DP-mechanism in Algorithm 1, the moments accountant of the sampled Gaussian mechanism correctly computes privacy loss with the noise scale of $z = \sigma / \mathbb{S}$ and central steps $T$, where $\mathbb{S} = C/(qN)$ and noise $\sigma$, clipping parameter $C$, probability of user selection $q$, and total number of users in the population $N$ are given in Algorithm 1.*

While we use the moments accountant and idealized uniform sampling in this work, one may instead use DP-FTRL (Kairouz et al., 2021a), or use sampling amongst available devices as discussed in Talwar et al. (2023). We expect similar results to be attainable using either of those approaches, potentially at the cost of a small constant overhead in the required population sizes.

Since we use large transformer ASR models, user-level DP may significantly reduce the utility of training ASR models even in the absence of FL because the noise may overpower the gradients (Xu et al., 2023a; Bassily et al., 2014). Our initial experiments confirmed this problem and, to alleviate this issue, we employ a simple yet effective remedy: per-layer clipping.

**Per-Layer Clipping**    Per-layer clipping was proposed by McMahan et al. (2018).However, the authors did not report a significant improvement in their setting of LSTM models for language. On the contrary, in this work, we demonstrate that in the context of FL with DP for large transformer models, per-layer clipping is essential as it mitigates the imbalance of gradients across different layers in the attention blocks. Formally, we change the global clipping of clients' deltas from Algorithm 1, Step 12, to per-layer clipping $\text{clip}_{layer}(\boldsymbol{g}, C)$ defined as follows:

**Definition 3** *Per-layer clipping: Let the model gradient be $\boldsymbol{g} = (\boldsymbol{g}_1, \boldsymbol{g}_2, ..., \boldsymbol{g}_K)$, where $\boldsymbol{g}_i$ is the $i$-th layer gradient with total $K$ layers in the model. Then per-layer clipping function with clipping parameter $C = \sqrt{\sum_{i=1}^{K} C_i^2}$ is given as $\text{clip}_{layer}(\boldsymbol{g}, C) = (\tilde{\boldsymbol{g}}_1, \tilde{\boldsymbol{g}}_2, ..., \tilde{\boldsymbol{g}}_K)$ where $\tilde{\boldsymbol{g}}_i = \text{clip}(\boldsymbol{g}_i, C_i)$.*

In our experiments we use either $C_i = \frac{C}{\sqrt{K}}$ ("uniform" variant) or $C_i = C\sqrt{\frac{D_i}{\sum_{j=1}^{K} D_j}}$ ("dim" variant based on a layer dimension) where $D_i$ is the dimension of the $i$-th layer and $i = \overline{1, K}$, so that after per-layer clipping we still guarantee $\|\boldsymbol{\Delta}_n^t\|_2 \leq C$ needed for Theorem 1 in DP.

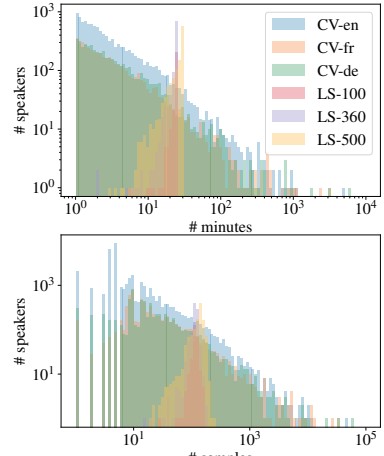

Table 1: Speaker statistics for LibriSpeech (LS) and Common Voice (CV) train sets and their subsets.

| Subset | # hours | # speakers | # minutes per speaker | | | |
|---|---|---|---|---|---|---|
| | | | mean | std | min | max |
| LS-100 | 100.6 | 251 | 24.1 | 2.7 | 5.5 | 25.2 |
| LS-360 | 363.6 | 921 | 23.7 | 3.2 | 1.9 | 25.3 |
| LS-500 | 496.9 | 1,166 | 25.6 | 5.9 | 3.0 | 30.3 |
| LS-860 | 860.5 | 2,087 | 24.7 | 5.1 | 1.9 | 30.3 |
| LS-960 | 961.1 | 2,338 | 24.7 | 4.9 | 1.9 | 30.3 |
| CV-en-train | 1593.7 | 34,753 | 2.8 | 32.7 | 0.02 | 5,049.6 |
| CV-en-train-10 | 149.5 | 3,475 | 2.6 | 17.3 | 0.03 | 755.1 |
| CV-en-train-90 | 1444.2 | 31,278 | 2.8 | 34.0 | 0.02 | 5,049.6 |
| CV-en-train-05 | 79.5 | 1,737 | 2.7 | 15.8 | 0.03 | 508.3 |
| CV-en-train-95 | 1514.2 | 33,016 | 2.7 | 33.4 | 0.02 | 5,049.6 |
| CV-fr-train | 727.9 | 6,856 | 6.4 | 57.2 | 0.04 | 3081.2 |
| CV-fr-train-10 | 47.6 | 685 | 4.2 | 13.6 | 0.07 | 235.1 |
| CV-fr-train-90 | 680.3 | 6,171 | 6.6 | 60.2 | 0.04 | 3081.2 |
| CV-de-train | 852.8 | 7,127 | 7.2 | 89.2 | 0.03 | 6249.9 |
| CV-de-train-10 | 52.2 | 712 | 4.4 | 11.4 | 0.04 | 120.8 |
| CV-de-train-90 | 800.6 | 6,415 | 7.5 | 94.0 | 0.03 | 6249.9 |

Figure 1: Speakers distribution in LibriSpeech (LS) and Common Voice (CV) train data: number of minutes per speaker (top) and number of samples per speaker (bottom).

## 4 EMPIRICAL SETUP

We perform all experiments using two datasets of audio-transcription pairs: LibriSpeech (Panayotov et al., 2015) and Common Voice v13.0 (Ardila et al., 2020). These two datasets are read speech but differ in other properties, like data diversity, noise conditions, speaker variation, and speaker distribution. We not only present results with English locale in LibriSpeech and Common Voice v13.0 but also complement them with results on French and German locale from Common Voice v13.0.

**LibriSpeech (LS)** We use *train-clean-100* (*LS-100*), *train-clean-360* (*LS-360*) and *train-other-500* (*LS-500*) as training data. *LS-960* is the union of *LS-100*, *LS-360* and *LS-500*. *LS-860* is the union of *LS-360* and *LS-500*. We use standard validation (*dev-clean* and *dev-other*) and test (*test-clean* and *test-other*) sets. For all data the original 16kHz sampling rate is maintained.

**Common Voice (CV), v13.0 (English, German and French)** We use the train, validation and test sets provided in the dataset. In addition, we split the training data using a specific percentage of users to train a seed model only and the rest of users for FL training: e.g., we create *CV-en-train-10(-5)* by selecting all the data for a randomly chosen 10% (5%) of the users from *CV-en-train* and we denote the remaining data by *CV-en-train-90(-95)*. Audio data are downsampled to 16kHz sampling rate.

Validation data are used to tune all hyper-parameters and to select the best models based on the word error rate (WER), while the test sets are used only for final evaluation. Data statistics on the number of speakers and the number of minutes per speaker are given in Table 1 and Figure 1 for both LS and CV datasets and their subsets. The statistics show that CV data are much more heterogeneous than LS as pointed out by Gao et al. (2022). CV data thus enable a more realistic scenario for testing FL and FL with DP. The most realistic scenario for FL uses a small central dataset to train a seed model (e.g. *LS-100*), and a larger dataset from a different distribution for FL training (e.g. *CV-en-train*).

**Token Set** Likhomanenko et al. (2021b) showed that for data from different domains, it is better to use character tokens. Since in this paper we consider settings with data from different domains, the token set used in all our experiments is composed of English characters (a-z), augmented with a word boundary token, hyphen and apostrophe, resulting in a total of 29 characters. For French and German, common non-English characters are included as well.

**Central Training** We use standard feature extraction for audio (Synnaeve et al., 2020; Gulati et al., 2020) by computing log-mel filterbanks with 80 coefficients with a 25ms sliding window and 10ms stride length, later normalized to zero mean and unit variance for each input sequence. We employ a vanilla encoder-based transformer model trained with the CTC loss (Graves et al., 2006)[2]. We start our experimentation with the state-of-the-art model on *LS-100* from Likhomanenko et al. (2021a): (i) 1D convolution to perform striding (kernel of 7 with stride of 3); (ii) a transformer encoder with

---

[2]We focus only on a CTC model as it contains only the encoder part, is simpler to train in practice compared to Seq2Seq or Transducer models, and is less likely to over-fit to the language model (Synnaeve et al., 2020).

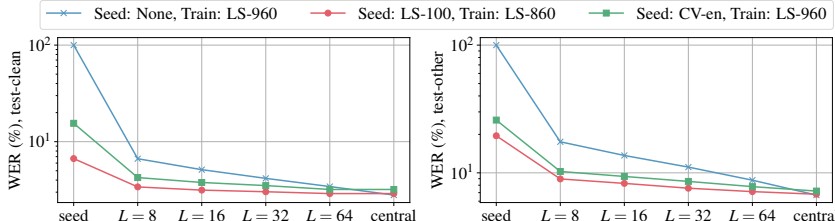

Figure 2: Comparison of WERs between central training and FL, and the impact of the cohort size $L$ and seed models for models trained on LS. We use exponential decay for central LR starting at $t = 1,000$, decay rate 0.6, and transition steps 500 (w/o seed model) or 250 (w/ seed model) with $T = 2k$ total central steps and 10 local epochs. Local (central) LR is 0.4 (0.006) (w/o seed model) or 0.2 (0.003) (w/ seed model). Detailed results can be found in Appendix C.1, Table 3.

36 layers, post-LayerNorm, 4 attention heads, an embedding dimension of 768, an MLP dimension of 3072, a dropout and layer drop (Fan et al., 2020) of 0.3; and (iii) a linear layer to map to the target vocabulary. The resulting model has 255M trainable parameters. We use SpecAugment (Park et al., 2019) and clip all gradients during training to have a norm of at most 1 (see Appendix B and C.5 for a discussion). We found it hard to switch to FL from central training when post-LayerNorm was used (similar issues were reported by Zhai et al. (2023)); see Appendix C.2 for an analysis. Following Zhai et al. (2023) we thus do central training with pre-LayerNorm (also used in FL), LARS (You et al., 2017), and relatively high (0.5) learning rate (LR) without any warmup and with step-wise decay to simplify the recipe and have stable training while maintaining the performance. We train on 8 GPUs (A100 80GB), and use a dynamic batch size of $\sim 240s$ audio per GPU.

**Federated Training** We simulate FL by considering every speaker in the data as a separate user. In most experiments, SGD (Sutskever et al., 2013) with constant LR is used as the local optimizer (configuration of central training made it possible) and LAMB (You et al., 2017) is used as the central optimizer. We found this combination most robust across all the scenarios (see Appendix C.3 for an analysis). The central (server) LR is constant with further exponential decay unless noted otherwise. All dropout and layer drop are fixed to 0.3, gradient clipping is set to 1 for each client. We train each client with a dynamic batch size of total 120s audio (CV) or 360s audio (LS). Unless noted otherwise, we restrict the number of central steps to 2k. Although most simulations would further improve after 2k steps, the per-step latency typically limits the number of iterations in practical private FL systems to this range (Xu et al., 2023b; Azam et al., 2023). Additionally, to keep simple and robust training recipes, we do not do extensive hyper-parameters search. After finding the best configuration on one training setup we apply exactly the same hyper-parameters to the rest of experiments.

## 5 RESULTS

### 5.1 IMPACT OF SEED MODELS AND COHORT SIZE

In Figures 2 and 3 we show that initializing FL with seed models significantly improves performance for both LS and CV (English, German, French), even with domain shift for the data on which the seed model is trained (e.g, using LS seed model for CV and vice-versa). Using seed model initialization for FL, we can achieve nearly optimal FL models within 2k central steps and moderate cohort sizes ($\geq 64$ for LS and $\geq 128$ for CV): their performance is on par with the centrally trained models. Larger cohorts consistently improve the outcomes within 2k central steps; this is expected as increasing the cohort size directly increases the amount of data processed during the training. Even without seed model initialization, the resulting FL models are competitive with centrally trained models given a large enough cohort size.

Increasing the amount of data for seed model training improves the trained FL models regardless of whether the data come from the same domain or not (e.g. compare *CV-en-train-05* seed vs. *CV-en-train-10* seed or *LS-100* seed vs. *LS-960* seed on *CV-en-train* in Figure 3). In fact, the use of seed models trained on considerably more data from another domain can outperform the use of seed models trained on less data from the same domain: the results on *CV-en-train* with a *LS-960* seed model are better than the results with a *CV-en-train-10* seed model on *CV-en-train-90*. We show

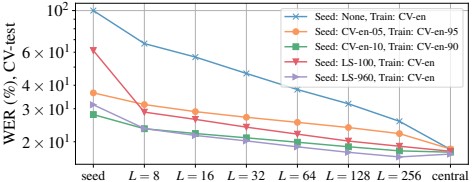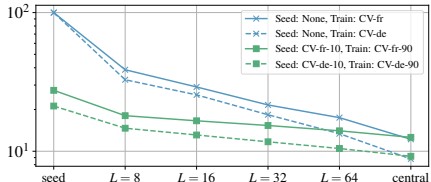

Figure 3: Comparison of WERs between central training and FL, and the impact of the cohort size $L$ and seed models for models trained on CV: English (left) and French/German (right). We use exponential decay for central LR starting at $t = 1,000$ (w/o seed model) or 750 (w/ seed model), decay rate 0.6, and transition steps 500 (w/o seed model) or 750 (w/ seed model) with $T = 2k$ total central steps and 10 local epochs. Local (central) LR is 0.4 (0.006) (w/o seed model) or 0.2 (0.002) (w/ seed model). Detailed results can be found in Appendix C.1, Tables 4 and 10.

additional results in Appendix C.8, Table 15, with the seed models trained for smaller number of steps: the better the seed models, the better the final FL models. The gap between FL models with different seed models decreases as the cohort size increases because the latter directly increases the amount of data processed during FL training.

In Figures 2 and 3 we use the same LR and LR decay schedule for all seed models regardless of the cohort size or the data used to train the seed model. However, optimal hyper-parameters like the learning rate are likely to depend on the quality of the seed model and cohort size. Thus, the results could likely be further improved by tuning the LR and its decay schedule for each cohort size and seed model separately. Furthermore, we can improve models by longer training exceeding 2k central steps as shown in ablations in Appendix C.7, Table 14.

To demonstrate robustness of found hyper-parameters and observed results in Figure 3 (left), we applied the **exact same training configuration** to train FL models on CV French and German data. *We confirm in Figure 3 (right) that the training configuration found on English data is robust: similar trends and results hold for French and German.*

## 5.2 IMPACT OF DATA HETEROGENEITY

Numerous prior works argued that data heterogeneity poses a challenge for FL (Li et al., 2020; Wang et al., 2020). Figure 4 shows that distributing data uniformly and randomly across users indeed improves performance for all datasets, cohort sizes and seed models. Since for LS, all clients have a dataset of approximately the same duration and we use dynamic batching, this is unlikely to be due to the differences in the amount of data between clients. The impact of using randomized, i.i.d. data decreases with increasing cohort size. Results in Figure 4 suggest that algorithms such as FedProx (Li et al., 2020), ProxSkip (Mishchenko et al., 2022), and SCAFFOLD (Karimireddy et al., 2020) could further improve FL performance. We evaluated FedProx and it indeed in many cases (for larger cohorts and w/o seed model) improves FL models, though marginally (see Appendix C.6, Table 13).

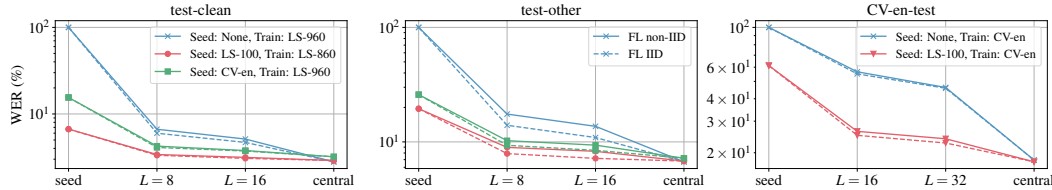

Figure 4: Impact of randomizing the distribution of data across users for LS (left, middle) and CV (right) measured by WER. Parameter settings are described in Figure 2 for LS and Figure 3 for CV. While the original training data are non-IID (solid), IID (dashed) versions of *LS-960*, *LS-860* and *CV-en-train* are created by choosing a user id uniformly and randomly from the set of user ids for each data point in the corresponding dataset. Detailed numbers are in Appendix C.1, Tables 5 and 6.

Table 2: Results for FL with DP and a model pre-trained on *LS-100* ($\sim$100h) used as central data and afterwards fine-tuned on *CV-en-train* ($\sim$1.6k hours) used as clients data. We report added noise $\mathcal{N}(0, I\sigma^2 Nq)$ per client and CV dev and test WERs (%) for two clipping variants with clipping bound $C$: global and per-layer "uniform" ("dim"). The total number of users is $N$, the expected number of users sampled per central step is $L = qN$, and the number of central steps is $T$. We set $\delta = 10^{-9}$ and report $\epsilon$ for which $(\epsilon, \delta)$-DP holds for given $L$ and $N$ using the moments accountant of Abadi et al. (2016). For scaling $L$ and $N$ where it is practically intractable to run model training (marked "-"), we extrapolate $(\epsilon, \delta)$-DP assuming the training dynamic remains unchanged and thus similar WER could be obtained. Central training gives 14.7%/17.8% WER on dev/test. Extended results are given in Appendix D and in Table 16. $\epsilon$ should be below 10 to be practically useful (marked with blue).

| $z$ | $\sigma$ $\cdot 10^{-8}$ | $C$ | $L$ | $N$ | $q = L/N$ | $T$ | $\epsilon$ | Renyi order | global clipping | | per-layer clipping: uniform (dim) | |
|---|---|---|---|---|---|---|---|---|---|---|---|---|
| | | | | | | | | | dev WER | test WER | dev WER | test WER |
| - | - | - | 0 | 34,753 | 0 | 0 | 0 | - | 54.7 | 61.2 | 54.7 | 61.2 |
| 0.03072 | 30.0 | 0.01 | 1,024 | 34,753 | 0.0295 | 2,006 | $1.1 \cdot 10^6$ | 1.1 | - | - | 25.2 (24.2) | 29.3 (28.2) |
| 0.3072 | 30.0 | 0.01 | 10,240 | 347,530 | 0.0295 | 2,006 | $3.7 \cdot 10^2$ | 1.1 | - | - | - | - |
| 1.536 | 30.0 | 0.01 | 51,200 | 1,737,650 | 0.0295 | 2,006 | $6.5 \cdot 10^0$ | 7.0 | - | - | - | - |
| 0.02048 | 20.0 | 0.01 | 1,024 | 34,753 | 0.0295 | 2,006 | $2.6 \cdot 10^6$ | 1.1 | - | - | 23.7 (22.6) | 27.6 (26.5) |
| 1.024 | 20.0 | 0.01 | 51,200 | 1,737,650 | 0.0295 | 2,006 | $1.3 \cdot 10^0$ | 4.0 | - | - | - | - |
| 2.048 | 20.0 | 0.01 | 102,400 | 3,475,300 | 0.0295 | 2,006 | $4.5 \cdot 10^0$ | 9.0 | - | - | - | - |
| 0.01024 | 10.0 | 0.01 | 1,024 | 34,753 | 0.0295 | 2,006 | $1.1 \cdot 10^7$ | 1.1 | - | - | **21.3 (20.1)** | **25.0 (23.7)** |
| 0.512 | 10.0 | 0.01 | 51,200 | 1,737,650 | 0.0295 | 2,006 | $7.2 \cdot 10^1$ | 1.5 | - | - | - | - |
| 1.024 | 10.0 | 0.01 | 102,400 | 3,475,300 | 0.0295 | 2,006 | $1.3 \cdot 10^1$ | 4.0 | - | - | - | - |
| 2.048 | 10.0 | 0.01 | 204,800 | 6,950,600 | 0.0295 | 2,006 | $4.5 \cdot 10^0$ | 9.0 | - | - | - | - |
| 0.003072 | 3.0 | 0.01 | 1,024 | 34,753 | 0.0295 | 2,006 | $1.2 \cdot 10^8$ | 1.1 | 27.0 | 31.1 | **17.9 (17.1)** | **21.2 (20.4)** |
| 0.3072 | 3.0 | 0.01 | 102,400 | 3,475,300 | 0.0295 | 2,006 | $3.7 \cdot 10^2$ | 1.1 | - | - | - | - |
| 0.6144 | 3.0 | 0.01 | 204,800 | 6,950,600 | 0.0295 | 2,006 | $4.2 \cdot 10^1$ | 2.0 | - | - | - | - |
| 0.6144 | 3.0 | 0.01 | 204,800 | 69,506,000 | 0.00295 | 2,034 | $7.2 \cdot 10^0$ | 3.0 | - | - | - | - |
| 0.6144 | 3.0 | 0.01 | 204,800 | 695,060,000 | 0.000295 | 3,390 | $3.7 \cdot 10^0$ | 6.0 | - | - | - | - |
| 0.001024 | 1.0 | 0.01 | 1,024 | 34,753 | 0.0295 | 2,006 | $1.1 \cdot 10^9$ | 1.1 | 22.9 | 26.7 | 16.2 (16.0) | 19.5 (19.3) |
| 0.2048 | 1.0 | 0.01 | 204,800 | 6,950,600 | 0.0295 | 2,006 | $1.1 \cdot 10^3$ | 1.1 | - | - | - | - |
| 0.2048 | 1.0 | 0.01 | 204,800 | 69,506,000 | 0.00295 | 2,034 | $2.7 \cdot 10^2$ | 1.1 | - | - | - | - |
| 0.2048 | 1.0 | 0.01 | 204,800 | 695,060,000 | 0.000295 | 3,390 | $9.4 \cdot 10^1$ | 1.3 | - | - | - | - |
| - | 0 | 0.01 | 1,024 | 34,753 | 0.0295 | 2,000 | inf | - | 15.7 | 18.9 | 15.9 | 19.1 |
| - | 0 | 1.0 | 1,024 | 34,753 | 0.0295 | 2,000 | inf | - | 15.7 | 18.9 | 15.9 | 19.1 |

## 5.3 FEDERATED LEARNING WITH DIFFERENTIAL PRIVACY

For FL with DP we consider a setting close to the real-world scenario: *LS-100* is used as central data to train a seed model (without DP); *CV-en-train* is considered as clients' data on which the seed model is trained afterwards using FL. In this setting (i) the clients' data are $\sim$16 times bigger than the server data and (ii) there is a domain shift in clients' data.

As discussed in Section 3, DP is challenging for larger models due to their size. To make the model training more resistant to noise, we need to increase the cohort size, e.g. in recent work Chatzidakis et al. (2023) used 150k cohort size for FL with DP. We take exactly the same setup as in Figure 3 with the data *CV-en-train* and the seed model trained on *LS-100*. First we scale the FL training to the cohort size of 1024; to mitigate the resulting increase in the computational cost of the training, we switch from 10 local epochs to 10 local steps (see Appendix D, all other hyper-parameters stay the same). Increasing the cohort size further closes the gap with the central baseline. Second, we use and vary the clipping bound $C$ applied to clients' deltas (global clipping) without adding DP noise yet. Interestingly, although the average norm of clients' deltas is 0.7 (see Appendix D, Figure 8), clients' deltas can be clipped with the clipping bound as low as $C = 10^{-8}$ without any impact on the quality of the final models. We attribute this partially to LARS component of LAMB, which re-normalizes the gradients. For further experiments, we set $C = 10^{-2}$ to prevent precision errors. Finally, we add different levels of noise $\sigma$ to every client's delta before averaging the deltas across clients.

All results are given in Table 2. We estimate $(\epsilon, \delta)$-DP by the moments accountant of Abadi et al. (2016) for every level of noise, number of clients $N$, clients sampling $q$, clients' deltas clipping $C$, and number of central training steps $T$. Using FL with DP on clients data, we can improve over the poor performing *LS-100* seed model due to limited server data and their domain shift: WER is reduced from 61.2% to 31.1% with $\sigma = 3 \cdot 10^{-8}$ and $(7.2, 10^{-9})$-DP assuming the training effectiveness (WER) remains the same if we extrapolate to $\sim$70M clients with the cohort size of $\sim$200k. Lowering the DP noise $\sigma$ decreases model's WER; however, DP guarantees become impractical even if we scale the number of clients and cohort size.

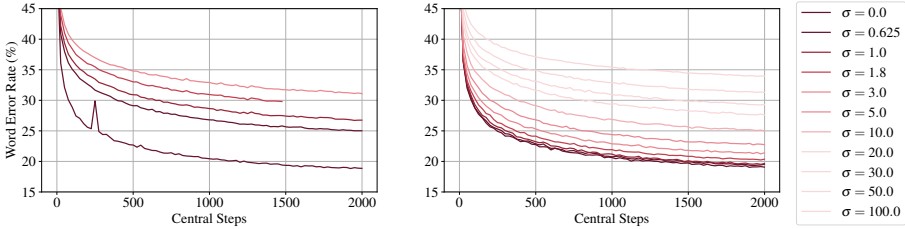

Figure 5: Client delta norms computed per layer in the model. We average statistics across all clients and central steps, and plot the mean and standard deviation. The model is trained with $\sigma = 3 \cdot 10^{-8}$ and global clients' deltas clipping $C = 10^{-2}$ (Algorithm 1). Transformer block consists of attention parameters (wqkv and wf) with LayerNorm (ln1), and MLP (w1 and w2) with LayerNorm (ln2).

Figure 6: Word error rate (WER) measured on *CV-en-test* set for different values of DP noise $\sigma$ (scale is set to $10^{-8}$). We apply clipping of $10^{-2}$ either globally (left, Algorithm 1) or per-layer (right, Definition 3, "uniform") with $T = 2k$ central steps, and $L$=1,024 cohort size. The seed model is trained on *LS-100*. A detailed plot is shown in Appendix D, Figure 12.

In Figure 5, we analyse the clients' deltas by computing model's per-layer deltas norm. We highlight that the norms are imbalanced across different transformer layers and also across different types of parameters: (i) first transformer layers have a larger deltas norm magnitude; and (ii) delta norms for attention parameters are an order of magnitude lower than those for LayerNorms. This explains why FL performed the best with LAMB and LARS central optimizers, which adapt LR per layer.

*To avoid the DP noise $\sigma$ dominating the attention layers and slowing down the convergence*, we propose to apply per-layer clipping (Definition 3) instead of global clipping of Algorithm 1. By applying per-layer clipping, we significantly improve model convergence (see Figure 6): with the same DP noise $\sigma = 3 \cdot 10^{-8}$ we are able to closely match the model trained without DP noise ($\sigma = 0$) with only a small WER degradation (from 19.1% to 21.2% WER) while guaranteeing $(7.2, 10^{-9})$-DP assuming the training effectiveness remains the same if we scale to $\sim$70M clients with the cohort size of $\sim$200k. Moreover, we can now increase DP noise up to $\sigma = 10^{-7}$ getting 23.7% WER with $(4.5, 10^{-9})$-DP by scaling only to $\sim$7M clients with the cohort size of $\sim$200k (see Table 2). The latter is a realistic scenario even for mid/low resource languages. We can further reduce WER by $\sim$1% for the same $(\epsilon, \delta)$-DP guarantee if we apply per-layer clipping based on the layer dimension (see Table 2).

## 6 CONCLUSION

ASR provides a valuable benchmark for (private) federated learning (FL). Datasets in this domain are large, separated by users, and represent heterogeneity of the kind often seen in real private FL settings. With the possible exception of language modeling, benchmarks typically used in the private FL research community do not satisfy these properties, making them less suitable for deriving conclusions that would translate to practical deployments. In this paper, we have argued that e.g. the task of adapting a model trained (centrally) on LibriSpeech data to work on Common Voice data in the federated setting is an excellent candidate for benchmarking FL and FL with differential privacy (DP). With a *practical* number of central aggregations, we have been able to train large transformer-based FL models that are nearly optimal even with heterogeneous data, a seed model from another domain, or no pre-trained seed model. Finally, we have showed that FL with DP for the above benchmark is not straightforward and entails several challenges. We propose to overcome some of them by reviving per-layer clipping, which allows us to achieve user-level $(7.2, 10^{-9})$-DP (resp. $(4.5, 10^{-9})$-DP) with a 1.3% (resp. 4.6%) absolute drop in the word error rate for extrapolation to high (resp. low) population scale.

ETHICS STATEMENT

For all experiments we use publicly available datasets for research: LibriSpeech (CC BY 4.0) and Common Voice v13.0 (CC BY-SA 3.0). In the paper, we aim at understanding the behaviour of large transformer models in federated learning with differential privacy. This is a step towards developing private federated learning in the context of speech recognition to provide strong guarantees on the user privacy.

REPRODUCIBILITY STATEMENT

For all experiments we use publicly available datasets for research: LibriSpeech (CC BY 4.0) and Common Voice v13.0 (CC BY-SA 3.0). Data processing is described in the main body of the paper. We also plan to release the code with the main experiments. However, we tried as much as possible to describe all configurations, training details, ablations, and our procedure of selecting hyper-parameters throughout the paper and in Appendix. We also provide important discussions on different aspects of the empirical results as well as detailed plots of various characteristics tracked during training in Appendix.

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

# A    DISCUSSION

## A.1    NEED FOR PRIVATE FEDERATED LEARNING

In Section 1 we discussed that FL on its own does not guarantee user privacy. For example, Boenisch et al. (2023) showed that the gradients sent to the server can be used to reconstruct the original training images and text. Carlini et al. (2023) showed that a model can memorize specific pieces of data that can be reconstructed using only the model itself. In the context of ASR, Tomashenko et al. (2022) developed two attacks that aim to infer speaker identity from the model updates without access to the actual users' audio data. Kröger et al. (2020) showed that audio data reveal information about the content but they can also be used to derive other pieces of sensitive information including biometric identity, physical traits, geographical origin, emotions, level of intoxication, age, gender and health.

These and many other works emphasize the necessity of developing private FL with strong guarantees on the user privacy. In this paper, we focus on providing first insights for private FL with DP for ASR.

## A.2    WHY DO WE STUDY LARGER MODELS FOR FL AND DP?

As discussed in Section 1, we focus on the model size of 250M parameters. Prior works in FL with DP primarily focused on studying models of up to 30M parameters, justifying the use of smaller models by communication and training costs associated with the model size and the difficulty of training reasonable models with DP because the impact of noise scales with the model size. However, Li et al. (2022b;a) showed that it is possible to (centrally) fine-tune large language models with hundreds of millions of parameters with DP and DP impact does not prevent efficient training if gradients are low rank.

Our main reason to focus our study on larger models for both FL and DP is the observation that larger models are simpler to train in practice. It is a hard and open problem to efficiently train small models that perform the same or better than models obtained for example by distillation of large models into smaller models (Stanton et al., 2021). To disentangle the ability to train small models efficiently from the problem of matching central training with FL and FL with DP, we study larger models. Our results give a hint that the gap that existed between FL and central models could be related to the absence of proper training recipes for smaller models.

One could argue that current model sizes are huge in the era of large language models, and different techniques, like LoRA (Hu et al., 2022), could be used to reduce training time on clients as well as communication costs. This was done for example by Xu et al. (2023a) who used partial and low-rank model updates to train large language models with private FL. However, we believe that first we need to train competitive baseline models from scratch or from out-of-domain seed models, and understand their behaviour and limits.

## A.3    CLIPPING AND ADAPTIVE OPTIMIZERS

Zhang et al. (2022a) investigated how clipping fights data heterogeneity in FL. As discussed in Section 3, clipping is also an essential part of DP. Interestingly, to be able to train transformer models, we must use clipping too, and thus the recipes used for transformers are aligned with FL with DP. In Appendix D Figure 8, we show that gradient clipping during local training leads to bounded norms of user deltas where the latter is necessary for DP. Without applying gradient clipping, the gradient norms would be huge already at the beginning of the training and even with LARS, pre-LayerNorm and central training we would not be able to train a reasonable model. Thus, it is extremely hard to disentangle any empirical results for transformers to understand how clipping helps the training for FL with DP.

Reddi et al. (2021) and Azam et al. (2023) showed that adaptive optimizers alleviate the issue of data heterogeneity for FL. At the same time it is hard to train transformer models without adaptive optimizers (Zhang et al., 2020; 2022a). This is yet another example of alignment between FL and central training of transformer models; a technique that helps alleviate data heterogeneity in FL is a must when training large transformer models even centrally.

### A.4 Fusion of ASR Model with a Language Model

To further improve WERs, ASR models can be combined with language models during inference. This can be done in various ways, e.g. using beam-search decoding for CTC models (Synnaeve et al., 2020; Likhomanenko et al., 2021b), or using shallow fusion (Toshniwal et al., 2018), cold fusion (Sriram et al., 2017), deep fusion (Gülçehre et al., 2015), and simple fusion (Stahlberg et al., 2018) for Seq2Seq or transducer-based models. In this paper, we leave the study on how a language model integration affects the final model performance as a future work. In the latter case, language models can also be trained using FL with DP (McMahan et al., 2018; Xu et al., 2023a;b).

### A.5 Conformer vs Transformer

Purposefully, we do not use the conformer architecture (Gulati et al., 2020) in the paper. In prior work by Kim et al. (2022), it was shown that, e.g., for CTC models both conformer and transformer architectures give similar results while conformer has fewer parameters. We focus on larger models to understand their behaviour. Moreover, vanilla transformers are still de facto a standard in other domains, while conformers were adopted only in speech recognition. Therefore, focusing on vanilla transformer models will broaden the impact of our findings for speech recognition on the FL and DP communities at large.

### A.6 Seed Models

Gao et al. (2022) trained seed models to initialize FL using a small fraction of speakers (117 speakers, or 2.8%, for French and 99 speakers, or 13.2%, for Italian) and used the rest of the data for FL training. Recent work (Berrebbi et al., 2023) showed that model quality depends on the number of speakers and the diversity of the training data: it is better to have more speakers with shorter total audio duration than to have fewer speakers with longer total audio duration.

Based on the recommendation of Berrebbi et al. (2023) to have at least 1k speakers in the training data, we randomly sampled 5% (English) or 10% (all languages) of speakers for the in-domain seed model training. This provided more than 1k users for training CV seed models for English. While for French the seed model is trained from only 685 users and for German the seed model is trained from only 712 users, we note that French and German languages are easier to train. Furthermore, for FL models training on CV (English) we use a seed model trained on *LS-100* that has only 251 speakers; however, *LS-100* has over 100 hours of audio, which is approximately 6.3% of the total audio in CV.

Preliminary experiments showed that the seed model training on a subset of 5% speakers with the shortest total audio does not converge: even for English the subset contains less than 2 hours of audio, which is known to be hard for any E2E ASR model training. In contrast, if we take a subset of 5% speakers with the longest total audio as in Gao et al. (2022), a seed model is very well trained as then the dataset has more than 64% of total audio in the CV dataset for English language and training on the rest of the data brings little benefit. Thus, we found the subsets with minimum-duration or maximum-duration users to not be practical scenarios.

## B Central Models Training

**Data preprocessing** For CV English, transcriptions are normalized similarly as for LS by (i) lower casing; (ii) removing punctuation while preserving hyphen; and (iii) converting non-English characters into English ones with `unidecode`[3] package. For CV French and German, we do not remove non-English characters and we retain single quotes.

**Positional Embedding** To reduce model training time by a factor of approximately 2-3 and to reduce the memory footprint, we use CAPE positional embedding (Likhomanenko et al., 2021c) instead of relative positional embedding (Shaw et al., 2018); both models perform similarly.

---

[3] `https://pypi.org/project/Unidecode`.

Table 3: Results (WER %) on LS. All runs use exponential decay for central LR starting at iteration 1,000, decay rate 0.6, and transition steps 500 (w/o seed model) or 250 (w/ seed model). Local learning rate is 0.4 (w/o seed model) or 0.2 (w/ seed model). Central learning rate is 0.006 (w/o seed model) or 0.003 (w/ seed model). The number of central steps is $T = 2k$ and the number of local epochs is 10.

| Data | seed: None; train: LS-960 | | | | | | seed: LS-100; train: LS-860 | | | | | | seed: CV-en; train: LS-960 | | | | | |
|---|---|---|---|---|---|---|---|---|---|---|---|---|---|---|---|---|---|---|
| | seed | 8 | 16 | 32 | 64 | central | seed | 8 | 16 | 32 | 64 | central | seed | 8 | 16 | 32 | 64 | central |
| dev-clean | 100.0 | 6.6 | 4.8 | 4.0 | 3.3 | 2.7 | 6.2 | 3.3 | 3.1 | 2.9 | 2.7 | 2.7 | 16.5 | 4.0 | 3.6 | 3.3 | 2.9 | 3.1 |
| test-clean | 100.0 | 6.7 | 5.1 | 4.2 | 3.4 | 2.8 | 6.7 | 3.4 | 3.2 | 3.0 | 2.9 | 2.9 | 15.5 | 4.3 | 3.8 | 3.5 | 3.2 | 3.2 |
| dev-other | 100.0 | 17.2 | 13.5 | 11.1 | 8.8 | 6.7 | 19.2 | 9.4 | 8.5 | 8.1 | 7.7 | 6.9 | 25.2 | 10.5 | 9.6 | 8.8 | 8.1 | 7.5 |
| test-other | 100.0 | 17.5 | 13.7 | 11.1 | 8.8 | 6.8 | 19.5 | 9.0 | 8.3 | 7.6 | 7.1 | 6.8 | 25.9 | 10.3 | 9.4 | 8.6 | 7.8 | 7.2 |

Table 4: Results (WER %) on CV. We use exponential decay for central LR starting at $t = 1,000$ (w/o seed model) or $t = 750$ (w/ seed model), decay rate 0.6, and transition steps 500 (w/o seed model) or 750 (w/ seed model) with $T = 2k$ total central steps and 10 local epochs. Local (central) LR is 0.4 (0.006) (w/o seed model) or 0.2 (0.002) (w/ seed model).

| Seed | Data | Eval. | seed WER | cohort size WER | | | | | | central WER |
|---|---|---|---|---|---|---|---|---|---|---|
| | | | | 8 | 16 | 32 | 64 | 128 | 256 | |
| None | CV-en | dev | 100.0 | 62.9 | 51.9 | 41.3 | 32.9 | 27.2 | 21.3 | 15.1 |
| | | test | 100.0 | 66.7 | 56.5 | 46.3 | 38.0 | 31.9 | 25.7 | 18.2 |
| CV-en-05 | CV-en-95 | dev | 31.3 | 26.6 | 24.3 | 22.7 | 21.2 | 19.8 | 18.2 | 15.2 |
| | | test | 36.4 | 31.6 | 28.9 | 27.0 | 25.4 | 23.8 | 22.1 | 18.3 |
| CV-en-10 | CV-en-90 | dev | 23.0 | 20.3 | 18.9 | 17.7 | 16.7 | 15.7 | 14.8 | 14.5 |
| | | test | 27.9 | 24.4 | 22.8 | 21.5 | 20.1 | 19.1 | 18.0 | 17.6 |
| LS-100 | CV-en | dev | 54.7 | 24.5 | 22.2 | 20.1 | 18.4 | 16.8 | 15.6 | 14.7 |
| | | test | 61.2 | 28.8 | 26.3 | 23.9 | 22.0 | 20.2 | 18.9 | 17.8 |
| LS-960 | CV-en | dev | 27.0 | 19.7 | 18.1 | 16.9 | 15.6 | 14.5 | 13.7 | 14.1 |
| | | test | 31.5 | 23.5 | 21.6 | 20.2 | 18.8 | 17.6 | 16.6 | 17.2 |

**SpecAugment** SpecAugment (Park et al., 2019) is activated from the very first step of training. Two frequency masks with frequency mask parameter $F = 30$, ten time masks with maximum time-mask ratio $p = 0.1$ and time mask parameter $T = 50$ are used; time warping is not used.

For all central models training, we use LARS optimizer with the learning rate of 0.5 (for models fine-tuned from seed models trained on *CV-*-train-10* we use 0.2) without a warmup period. Training is done for up to 300k-600k steps until full convergence with step-wise (by 2x) learning rate decay every 50k steps started at 40k-330k depending on the model.

## C   FEDERATED LEARNING WITHOUT DIFFERENTIAL PRIVACY

### C.1   DETAILED RESULTS FOR ENGLISH

Table 3 details the results for LS from Figure 2 and Table 4 details the results for CV from Figure 3. Table 5 details the results for randomized LS dataset (IID) from Figure 4 (left and middle). Table 6 details the results for randomized CV dataset (IID) from Figure 4 (right).

### C.2   IMPACT OF MODEL ARCHITECTURE ON FL PERFORMANCE IN ASR

Table 7 compares several model architectures for the trivial FL scenario with cohort size 1 and 64k central iterations on *LS-100*. Cohort size of 1 is impractical but it eliminates the impact of federated averaging. The learning rates and learning rate decay schedules are tuned for each architecture. During preliminary FL experiments we have observed that pre-LayerNorm models often perform better than post-LayerNorm ones. It is of note that without a linear central learning rate warmup, we were unable to train reasonable FL models with post-LayerNorm. Our experiments showed that FL models with pre-LayerNorm are easier to train, they do not require a central learning rate warmup,

Table 5: Impact of randomizing the distribution of data across users for LS measured by WER (%). Parameter settings are described in Table 3. While the original train data are non-IID, IID (columns with "IID") versions of *LS-960* and *LS-860* are created by choosing a user id uniformly and randomly from the set of user ids for each data point in the corresponding dataset.

| Data | seed: None; train: LS-960 | | | | | | seed: LS-100; train: LS-860 | | | | | | seed: CV-en; train: LS-960 | | | | | |
|---|---|---|---|---|---|---|---|---|---|---|---|---|---|---|---|---|---|---|
| | seed | 8 | 8-IID | 16 | 16-IID | central | seed | 8 | 8-IID | 16 | 16-IID | central | seed | 8 | 8-IID | 16 | 16-IID | central |
| dev-clean | 100.0 | 6.6 | 5.9 | 4.8 | 4.5 | 2.7 | 6.2 | 3.3 | 3.3 | 3.1 | 3.0 | 2.7 | 16.5 | 4.0 | 3.9 | 3.6 | 3.5 | 3.1 |
| test-clean | 100.0 | 6.7 | 6.0 | 5.1 | 4.7 | 6.7 | 2.8 | 3.4 | 3.3 | 3.2 | 3.1 | 2.9 | 15.5 | 4.3 | 4.1 | 3.8 | 3.7 | 3.2 |
| dev-other | 100.0 | 17.2 | 14.0 | 13.5 | 11.2 | 6.7 | 19.1 | 9.4 | 8.1 | 8.5 | 7.4 | 6.9 | 25.2 | 10.5 | 9.5 | 9.6 | 8.8 | 7.5 |
| test-other | 100.0 | 17.5 | 14.0 | 13.7 | 10.9 | 6.8 | 19.5 | 9.0 | 7.9 | 8.3 | 7.2 | 6.8 | 25.9 | 10.3 | 9.3 | 9.4 | 8.4 | 7.2 |

Table 6: Impact of randomizing the distribution of data across users for CV measured by WER (%). Parameter settings are described in Table 4. While the original train data are non-IID, the IID (columns with "IID") version of *CV-en-train* is created by choosing a user id uniformly and randomly from the set of user ids for each data point in the corresponding dataset.

| Seed | Data | Eval. | seed | cohort size WER | | | | central |
|---|---|---|---|---|---|---|---|---|
| | | | WER | 16 | 16-IID | 32 | 32-IID | WER |
| None | CV-en | dev | 100.0 | 51.9 | 50.2 | 41.3 | 40.9 | 15.1 |
| | | test | 100.0 | 56.5 | 55.0 | 46.3 | 45.8 | 18.2 |
| LS-100 | CV-en | dev | 54.7 | 22.2 | 21.1 | 20.1 | 19.1 | 14.7 |
| | | test | 61.2 | 26.3 | 25.0 | 23.9 | 22.7 | 17.8 |

Table 7: Comparison (WER, %) between pre-LayerNorm and post-LayerNorm architectures in transformer for trivial FL scenario with cohort size $L = 1$ and central steps $T = 64$k on *LS-100*. pre-LayerNorm models perform best and their training is robust with respect to hyper-parameters such as the learning schedule. Central models are trained according to Appendix B. FL models use exponential learning rate decay, LAMB as central and SGD as local optimizers.

| Model | Warmup | dev-clean | dev-other | test-clean | test-other |
|---|---|---|---|---|---|
| Central pre-LayerNorm | 0 | 5.9 | 18.9 | 6.4 | 19.2 |
| FL pre-LayerNorm | 0 | 5.6 | 17.7 | 5.9 | 17.9 |
| Central post-LayerNorm | 0 | 8.1 | 25.0 | 8.6 | 25.6 |
| FL post-LayerNorm | 1000 | 5.9 | 17.5 | 6.3 | 18.0 |

and they are generally more robust with respect to hyper-parameters. These observations are similar to prior works on transformers central training (Zhang et al., 2022a; Zhai et al., 2023). That is why we use the pre-LayerNorm configuration for all experiments in the paper. It is interesting that for this trivial FL scenario FL models outperforms centrally trained models. However, when we switch to larger *LS-960* dataset, this does not hold anymore.

## C.3 IMPACT OF SERVER OPTIMIZER ON FL PERFORMANCE IN ASR

Table 8 compares the LAMB optimizer (You et al., 2020) used as the central optimizer in all FL runs presented so far with Adam (Kingma & Ba, 2015) and LARS (You et al., 2017) on several configurations for *LS-960* dataset. The results on *LS-960* indicate that LAMB performs significantly better than LARS and Adam without a seed model, and it performs slightly better than LARS and Adam with a seed model. Adam performs slightly better than LARS.

Table 9 compares LAMB with Adam, AdaGrad (Duchi et al., 2011), LARS, and SGD (Sutskever et al., 2013) on several configurations for *CV-en* dataset. The results on CV show that without seed models, LAMB performs significantly better than all other optimizers but with seed models, LAMB is sometimes outperformed slightly by LARS and Adam. SGD, AdaGrad and Adam are outperformed by LAMB and LARS in almost all scenarios.

Table 8: Comparison (WER, %) of various server optimizers on *LS-960* with and without a seed model. For LAMB, the results and parameters are the same as those in Table 3 (note that these are sub-optimal because for simplicity we use the same learning rate and learning rate decay schedule for each configuration regardless of the cohort size and all runs with seed models use the same configuration). For all other optimizers, the central learning rate and the learning rate decay schedule are tuned separately for each combination of cohort size and seed model.

| Seed | Data | Cohort size | Central optimizer | LR | dev-clean $T=0$ | dev-clean $T=2k$ | test-clean $T=0$ | test-clean $T=2k$ | dev-other $T=0$ | dev-other $T=2k$ | test-other $T=0$ | test-other $T=2k$ |
|------|------|------|------|------|------|------|------|------|------|------|------|------|
| None | LS-960 | 8 | LAMB | 0.006 | 100.0 | 6.6 | 100.0 | 6.7 | 100.0 | 17.2 | 100.0 | 17.5 |
|      |        |   | LARS | 0.7 | 100.0 | 13.7 | 100.0 | 14.1 | 100.0 | 30.9 | 100.0 | 31.6 |
|      |        |   | Adam | 0.001 | 100.0 | 14.1 | 100.0 | 14.6 | 100.0 | 30.4 | 100.0 | 31.0 |
| None | LS-960 | 16 | LAMB | 0.006 | 100.0 | 4.8 | 100.0 | 5.1 | 100.0 | 13.5 | 100.0 | 13.7 |
|      |        |   | LARS | 0.7 | 100.0 | 10.5 | 100.0 | 11.0 | 100.0 | 25.9 | 100.0 | 25.9 |
|      |        |   | Adam | – | - | - | - | - | - | - | - | - |
| CV-en | LS-960 | 8 | LAMB | 0.003 | 16.5 | 4.0 | 15.5 | 4.3 | 25.2 | 10.5 | 25.9 | 10.3 |
|      |        |   | LARS | 1.2 | 16.5 | 4.2 | 15.5 | 4.4 | 25.2 | 10.6 | 25.9 | 10.6 |
|      |        |   | Adam | 0.012 | 16.5 | 4.3 | 15.5 | 4.3 | 25.2 | 10.7 | 25.9 | 10.5 |

Table 9: Comparison (WER, %) of various optimizers on *CV-en* with and wihout seed models. For LAMB, the results and parameters are the same as those in Table 4 (note that these are sub-optimal because for simplicity we use the same learning rate and learning rate decay schedule for each configuration regardless of the cohort size and all runs with seed models use the same configuration). For all other optimizers, the central learning rate and the learning rate decay schedule are tuned separately for each combination of cohort size and seed model.

| Seed | Data | Cohort size | Central optimizer | LR | dev $T=0$ | dev $T=2k$ | test $T=0$ | test $T=2k$ |
|------|------|------|------|------|------|------|------|------|
| None | CV-en | 8 | LAMB | 0.006 | 100.0 | 62.9 | 100.0 | 66.7 |
|      |       |   | LARS | 3.4 | 100.0 | 70.4 | 100.0 | 73.8 |
|      |       |   | Adam | 0.0005 | 100.0 | 68.9 | 100.0 | 72.2 |
|      |       |   | AdaGrad | 0.003 | 100.0 | 84.3 | 100.0 | 86.2 |
|      |       |   | SGD | 2.8 | 100.0 | 83.8 | 100.0 | 86.0 |
| None | CV-en | 16 | LAMB | 0.006 | 100.0 | 51.9 | 100.0 | 56.5 |
|      |       |   | LARS | 2.6 | 100.0 | 57.6 | 100.0 | 62.0 |
|      |       |   | Adam | 0.0005 | 100.0 | 57.7 | 100.0 | 62.1 |
|      |       |   | AdaGrad | 0.002 | 100.0 | 82.1 | 100.0 | 84.5 |
|      |       |   | SGD | 3.0 | 100.0 | 84.5 | 100.0 | 86.6 |
| CV-en-10 | CV-en-90 | 8 | LAMB | 0.002 | 23.0 | 19.4 | 27.9 | 23.5 |
|      |       |   | LARS | 0.3 | 23.0 | 18.7 | 27.9 | 22.6 |
|      |       |   | Adam | 0.004 | 23.0 | 18.9 | 27.9 | 22.9 |
|      |       |   | AdaGrad | 0.016 | 23.0 | 19.4 | 27.9 | 23.6 |
|      |       |   | SGD | 1.6 | 23.0 | 20.9 | 27.9 | 25.4 |
| CV-en-10 | CV-en-90 | 16 | LAMB | 0.002 | 23.0 | 18.3 | 27.9 | 22.1 |
|      |       |   | LARS | 0.4 | 23.0 | 18.0 | 27.9 | 21.8 |
|      |       |   | Adam | 0.006 | 23.0 | 18.3 | 27.9 | 22.1 |
|      |       |   | AdaGrad | 0.015 | 23.0 | 19.1 | 27.9 | 23.2 |
|      |       |   | SGD | 1.6 | 23.0 | 20.8 | 27.9 | 25.2 |
| CV-en-10 | CV-en-90 | 32 | LAMB | 0.002 | 23.0 | 17.3 | 27.9 | 21.0 |
|      |       |   | LARS | 0.6 | 23.0 | 17.3 | 27.9 | 20.9 |
|      |       |   | Adam | 0.006 | 23.0 | 17.5 | 27.9 | 21.1 |
| CV-en-10 | CV-en-90 | 64 | LAMB | 0.002 | 23.0 | 16.7 | 27.9 | 20.1 |
|      |       |   | LARS | 0.5 | 23.0 | 16.6 | 27.9 | 20.1 |
|      |       |   | Adam | 0.008 | 23.0 | 16.4 | 27.9 | 20.0 |

During hyper-parameter tuning, some adaptive optimizers (e.g., Adam) often became unstable and the training diverged, especially without a well performing seed model. Furthermore, the optimal parameters of these optimizers oftentimes vary significantly between, e.g., the cohort sizes, indicating that they are less robust than LAMB in our setting.

Table 10: Results (WER, %) on CV for English, French and German. Configurations are identical to those in Figure 3 and Table 4 regardless of the language.

| Seed | Data | Eval. | seed WER | cohort size WER | | | | | | central WER |
|------|------|-------|----------|------|------|------|------|------|------|------|
| | | | | 8 | 16 | 32 | 64 | 128 | 256 | |
| None | CV-en | dev | 100.0 | 62.9 | 51.9 | 41.3 | 32.9 | 27.2 | 21.3 | 15.1 |
| | | test | 100.0 | 66.7 | 56.5 | 46.3 | 38.0 | 31.9 | 25.7 | 18.2 |
| None | CV-fr | dev | 100.0 | 34.7 | 25.4 | 18.8 | 15.0 | - | - | 10.7 |
| | | test | 100.0 | 38.7 | 29.1 | 21.6 | 17.5 | - | - | 12.2 |
| None | CV-de | dev | 100.0 | 30.1 | 22.8 | 16.1 | 11.7 | - | - | 7.7 |
| | | test | 100.0 | 32.8 | 25.5 | 18.3 | 13.4 | - | - | 8.8 |
| CV-en-10 | CV-en-90 | dev | 23.0 | 20.3 | 18.9 | 17.7 | 16.7 | 15.7 | 14.8 | 14.5 |
| | | test | 27.9 | 24.4 | 22.8 | 21.5 | 20.1 | 19.1 | 18.0 | 17.6 |
| CV-fr-10 | CV-fr-90 | dev | 24.0 | 15.6 | 14.3 | 13.2 | 12.0 | - | - | 10.8 |
| | | test | 27.5 | 18.1 | 16.6 | 15.3 | 14.0 | - | - | 12.6 |
| CV-de-10 | CV-de-90 | dev | 18.6 | 12.8 | 11.4 | 10.2 | 9.1 | - | - | 8.1 |
| | | test | 21.2 | 14.7 | 13.1 | 11.7 | 10.5 | - | - | 9.2 |

The robustness of LAMB across all scenarios and its stability are the main reasons for choosing LAMB as the central optimizer for most of the experiments in the paper. However, the results in Table 9 suggest that some of the models could be further improved with more hyper-parameters tuning and choosing the best optimizer for each case. Also, Azam et al. (2023) showed that tuning other optimizer parameters, e.g. $\epsilon$ in Adam, can significantly improve FL model training for ASR. However, in this paper we restrict ourselves to tuning only the learning rate and learning rate schedule; the remaining parameters were set to their default values from `optax` library[4].

We have not completed an extensive evaluation of other optimizers for local training to keep it efficient (no state, no additional memory, no extra computations): SGD as a local optimizer is robust and efficient in our experiments. However, preliminary experiments show that LARS and LAMB are well suited candidates for replacing SGD as the local optimizer and will likely outperform SGD.

For completeness, here we provide more details on optimizer tuning. For both *LS-960* and *CV-en-train* without a seed model, we tuned the central LR for LAMB between 0.001 and 0.009, and the local LR for SGD from 0.2 to 0.6. We have done the same for one selected seed model for each dataset. Additionally, we tried several learning rate schedules, including constant rate, step decay, and exponential decay on several configurations. After the initial experiments, we chose one configuration for each dataset (LS, CV) without a seed model and one configuration for each dataset (LS, CV) with a seed model, and we ran the remaining experiments with the chosen configurations. The initial tuning was done on smaller cohort sizes. For other optimizers discussed in this section, we tuned the key parameters until a locally optimal value was found for central LR for each presented experiment, and we considered 4 variations of the exponential decay rate for each LR value.

### C.4 DETAILED RESULTS FOR CV FRENCH AND GERMAN

Table 10 shows the results of FL on CV for French and German languages, and for comparison it provides the corresponding results on CV for English. To demonstrate that the settings used for English language were robust, we did not tune any parameters for French and German, and simply used the exact same configuration that was used in the corresponding training on English language.

The results show that even though French and German have considerably smaller datasets, the training is apparently considerably easier and WERs are significantly smaller than for English whether or not a seed model is used. This is likely due to the degree of consistency between the orthography and phonology as was discussed for example in Borgwaldt et al. (2004); Borleffs et al. (2017); Ziegler et al. (1996); Sprenger-Charolles (2003); German and French have stronger orthography-to-phonology consistency than English. Furthermore, the results for French are considerably better than those presented by Gao et al. (2022). As French and German data are smaller, for the same cohort size

---

[4] https://optax.readthedocs.io/en/latest/

Table 11: Results (WER, %) on LS with and without SpecAugment ([Park et al., 2019](#)). Configurations are identical to those in Figure [2](#) and Table [3](#) except the SpecAugment schedule as noted in the Table.

| Seed | Data | SpecAugment | Cohort size | dev-clean | | test-clean | | dev-other | | test-other | |
|------|------|-------------|-------------|-----------|-----------|------------|-----------|-----------|-----------|------------|-----------|
| | | | | $T=0$ | $T=2k$ | $T=0$ | $T=2k$ | $T=0$ | $T=2k$ | $T=0$ | $T=2k$ |
| None | LS-960 | ✓ | 8 | 100.0 | 6.6 | 100.0 | 6.7 | 100.0 | 17.2 | 100.0 | 17.5 |
| None | LS-960 | ✗ | 8 | 100.0 | 6.6 | 100.0 | 6.8 | 100.0 | 19.3 | 100.0 | 19.4 |
| None | LS-960 | ✓ | 16 | 100.0 | 4.8 | 100.0 | 5.1 | 100.0 | 13.5 | 100.0 | 13.7 |
| None | LS-960 | ✗ | 16 | 100.0 | 5.4 | 100.0 | 5.5 | 100.0 | 16.5 | 100.0 | 16.5 |
| LS-100 | LS-860 | ✓ | 8 | 6.2 | 3.3 | 6.7 | 3.4 | 19.1 | 9.4 | 19.5 | 9.0 |
| LS-100 | LS-860 | ✗ | 8 | 6.2 | 3.3 | 6.7 | 3.3 | 19.2 | 10.2 | 19.5 | 9.8 |
| LS-100 | LS-860 | ✓ | 16 | 6.2 | 3.1 | 6.7 | 3.2 | 19.1 | 8.5 | 19.5 | 8.3 |
| LS-100 | LS-860 | ✗ | 16 | 6.2 | 3.2 | 6.7 | 3.2 | 19.1 | 9.9 | 19.5 | 9.5 |
| CV-en | LS-960 | ✓ | 8 | 16.6 | 4.0 | 15.5 | 4.3 | 25.2 | 10.5 | 25.9 | 10.3 |
| CV-en | LS-960 | ✗ | 8 | 16.6 | 3.8 | 15.5 | 4.1 | 25.2 | 11.5 | 25.9 | 11.2 |
| CV-en | LS-960 | ✓ | 16 | 16.6 | 3.6 | 15.5 | 3.8 | 25.2 | 9.6 | 25.9 | 9.4 |
| CV-en | LS-960 | ✗ | 16 | 16.6 | 3.5 | 15.5 | 3.8 | 25.2 | 10.9 | 25.9 | 10.6 |

Table 12: Results (WER, %) on CV with and without SpecAugment ([Park et al., 2019](#)). Configurations are identical to those in Figure [3](#) and Table [4](#) except the SpecAugment schedule as noted in the table.

| Seed | Data | SpecAugment | Cohort size | dev | | test | |
|------|------|-------------|-------------|-------|--------|-------|--------|
| | | | | $T=0$ | $T=2k$ | $T=0$ | $T=2k$ |
| None | CV-en | ✓ | 8 | 100.0 | 62.9 | 100.0 | 66.7 |
| None | CV-en | ✗ | 8 | 100.0 | 52.3 | 100.0 | 57.5 |
| None | CV-en | ✓ | 16 | 100.0 | 51.9 | 100.0 | 56.5 |
| None | CV-en | ✗ | 16 | 100.0 | 42.2 | 100.0 | 47.9 |
| None | CV-en | ✓ | 32 | 100.0 | 41.3 | 100.0 | 46.3 |
| None | CV-en | ✗ | 32 | 100.0 | 33.8 | 100.0 | 39.3 |
| CV-en-10 | CV-en-90 | ✓ | 8 | 23.0 | 20.3 | 27.9 | 24.4 |
| CV-en-10 | CV-en-90 | ✗ | 8 | 23.0 | 19.9 | 27.9 | 24.3 |
| CV-en-10 | CV-en-90 | ✓ | 16 | 23.0 | 18.9 | 27.9 | 22.8 |
| CV-en-10 | CV-en-90 | ✗ | 16 | 23.0 | 18.3 | 27.9 | 22.4 |
| CV-en-10 | CV-en-90 | ✓ | 32 | 23.0 | 17.7 | 27.9 | 21.5 |
| CV-en-10 | CV-en-90 | ✗ | 32 | 23.0 | 17.1 | 27.9 | 21.2 |
| LS-100 | CV-en | ✓ | 8 | 54.7 | 24.5 | 61.2 | 28.8 |
| LS-100 | CV-en | ✗ | 8 | 54.7 | 23.3 | 61.2 | 27.9 |
| LS-100 | CV-en | ✓ | 16 | 54.7 | 22.2 | 61.2 | 26.3 |
| LS-100 | CV-en | ✗ | 16 | 54.7 | 21.0 | 61.2 | 25.4 |
| LS-100 | CV-en | ✓ | 32 | 54.7 | 20.1 | 61.2 | 23.9 |
| LS-100 | CV-en | ✗ | 32 | 54.7 | 19.0 | 61.2 | 23.0 |
| LS-960 | CV-en | ✓ | 8 | 27.0 | 19.7 | 31.5 | 23.5 |
| LS-960 | CV-en | ✗ | 8 | 27.0 | 19.5 | 31.5 | 23.5 |
| LS-960 | CV-en | ✓ | 16 | 27.0 | 18.1 | 31.5 | 21.6 |
| LS-960 | CV-en | ✗ | 16 | 27.0 | 17.8 | 31.5 | 21.6 |
| LS-960 | CV-en | ✓ | 32 | 27.0 | 16.9 | 31.5 | 20.2 |
| LS-960 | CV-en | ✗ | 32 | 27.0 | 16.4 | 31.5 | 20.2 |

and central steps we do more epochs over data for French and German than for English CV. Thus, FL training can match the central training with smaller cohort size for both French and German compared to English. It is of note that French and German turn out to be easier also for FL with DP as shown in Appendix [D.6](#), Table [17](#).

## C.5  IMPACT OF SPECAUGMENT

In all experiments so far, we used SpecAugment ([Park et al., 2019](#)) activated from the very first step of training as was also common in most prior works. Table [11](#) shows the results with and without SpecAugment for several configurations analyzed in this paper on LS data. These results confirm that SpecAugment improves WER in all the cases.

However, Table 12 shows that SpecAugment appears to have a negative impact on the trained models for CV (English), especially for FL training without a seed model and small cohort sizes. This is surprising as prior works reported only improved results with SpecAugment for transformer models. These results also reveal another difference between benchmarks on LS and on CV.

It is possible that the results with SpecAugment on CV would improve if SpecAugment was turned on later in the training and its parameters were tuned for each scenario separately. Nonetheless, since in most scenarios SpecAugment either improved models or the differences were marginal, for simplicity, we use SpecAugment in all experiments in this paper.

### C.6  PERFORMANCE OF FEDPROX IN FL FOR ASR

Li et al. (2020) proposed FedProx to alleviate the impact of heterogeneous data on FL performance. Since the results presented earlier in Tables 5 and 6 suggested that heterogeneous data pose a challenge for FL also in our training, we also evaluate the impact of FedProx on model quality in ASR. For each configuration, we use FedProx with the regularization weight $\mu \in \{0.00001, 0.0001, 0.001, 0.01, 0.1, 1.0\}$ and chose the best result, as suggested by Li et al. (2020).

Table 13 presents the results of using FedProx in several scenarios on LS and CV datasets presented earlier in Tables 3 and 4. The results show FedProx improves model performance (WER is decreased) in 8 out of 10 training configurations tested, although in most cases the improvement is marginal. In one of the remaining cases there is no change and only in one case the results with FedProx are considerably worse than without it.

It is surprising how the optimal value of the key FedProx parameter $\mu$ changes considerably between the various scenarios. This suggests that it would make sense to evaluate adaptive $\mu$ as suggested by Li et al. (2020). We leave the use of adaptive $\mu$ and the investigation of how FedProx may improve FL training robustness (e.g. with respect to the number of local epochs or steps) for future work.

We also tried limiting the number of batches processed on each client (Wang et al., 2020) and normalizing users' deltas sent to the server (Charles et al., 2021) but neither approach improved the results. See Table 7 in Appendix D.2 for the results on limiting the number of batches (steps) processed for each client.

### C.7  EXTENDING THE NUMBER OF CENTRAL FL ITERATIONS

Table 14 shows that even though most FL models were stopped after 2k central steps, letting these models to train longer would further improve performance. However, due to the communication complexity for each central step, it is best to use a moderate number of central steps and maximize utility of the training by optimizing the parameters for local, on-device training, cohort sizes, and other key FL parameters.

### C.8  IMPACT OF UNDER-TRAINED SEED MODELS

Table 15 shows that choosing a better seed model improves performance across the board. Furthermore, the results presented previously in Table 4 show that using a seed model trained on more data improves FL performance, even if the data used to train seed models are from a different domain.

## D  FEDERATED LEARNING WITH DIFFERENTIAL PRIVACY

### D.1  DIFFERENTIAL PRIVACY NOISE DISCUSSION

There are different equivalent formulations how the noise can be added to the clients' deltas to introduce DP, which can cause confusion about the noise scale and how the moments accountant is applied:

1. Noise is added on the client level: $\mathbf{\Delta}^t = \frac{1}{L} \sum_{n=1}^{L} \left[ \mathbf{\Delta}_n^t + \mathcal{N}(0, I\sigma_{client}^2) \right]$. This is the definition that we use in this paper with $\sigma_{client} = \sigma \cdot \sqrt{L}$ and this definition was also used by Zhang et al. (2022a).

Table 13: Results (WER, %) of FedProx on selected configurations on LS (top) and CV (English) (bottom) datasets. All parameters except for FedProx $\mu$ are identical to those in Tables 3 and 4. Parameter $\mu \in \{0.00001, 0.0001, 0.001, 0.01, 0.1, 1.0\}$ is tuned separately for every case and the best result is provided for each base configuration.

| Seed | Data | fedprox $\mu$ | Cohort size | dev-clean | | test-clean | | dev-other | | test-other | |
|------|------|------|------|------|------|------|------|------|------|------|------|
| | | | | $T=0$ | $T=2$k | $T=0$ | $T=2$k | $T=0$ | $T=2$k | $T=0$ | $T=2$k |
| None | LS-960 | 0 | 8 | 100.0 | 6.6 | 100.0 | 6.7 | 100.0 | 17.2 | 100.0 | 17.5 |
| None | LS-960 | 0.1 | 8 | 100.0 | 6.4 | 100.0 | 6.7 | 100.0 | 17.5 | 100.0 | 17.5 |
| None | LS-960 | 0 | 16 | 100.0 | 4.8 | 100.0 | 5.1 | 100.0 | 13.5 | 100.0 | 13.7 |
| None | LS-960 | 0.1 | 16 | 100.0 | 4.9 | 100.0 | 5.1 | 100.0 | 13.4 | 100.0 | 13.5 |
| LS-100 | LS-860 | 0 | 8 | 6.2 | 3.3 | 6.7 | 3.4 | 19.1 | 9.4 | 19.5 | 9.0 |
| LS-100 | LS-860 | 0.0001 | 8 | 6.2 | 3.3 | 6.7 | 3.5 | 19.1 | 9.3 | 19.5 | 9.0 |
| LS-100 | LS-860 | 0 | 16 | 6.2 | 3.1 | 6.7 | 3.2 | 19.1 | 8.5 | 19.5 | 8.3 |
| LS-100 | LS-860 | 1.0 | 16 | 6.2 | 3.0 | 6.7 | 3.2 | 19.1 | 8.6 | 19.5 | 8.3 |

| Seed | Data | fedprox $\mu$ | Cohort size | Central LR | dev | | test | |
|------|------|------|------|------|------|------|------|------|
| | | | | | $T=0$ | $T=2$k | $T=0$ | $T=2$k |
| None | CV-en | 0 | 8 | 0.006 | 100.0 | 62.9 | 100.0 | 66.7 |
| None | CV-en | 0.01 | 8 | 0.006 | 100.0 | 63.4 | 100.0 | 67.4 |
| None | CV-en | 0 | 16 | 0.006 | 100.0 | 51.9 | 100.0 | 56.5 |
| None | CV-en | 0.0001 | 16 | 0.006 | 100.0 | 51.0 | 100.0 | 55.8 |
| None | CV-en | 0 | 32 | 0.006 | 100.0 | 41.3 | 100.0 | 46.3 |
| None | CV-en | 0.0001 | 32 | 0.006 | 100.0 | 40.0 | 100.0 | 44.9 |
| LS-100 | CV-en | 0 | 8 | 0.002 | 54.7 | 24.5 | 61.2 | 28.8 |
| LS-100 | CV-en | 0.1 | 8 | 0.002 | 54.7 | 24.3 | 61.2 | 28.7 |
| LS-100 | CV-en | 0 | 16 | 0.002 | 54.7 | 22.2 | 61.2 | 26.3 |
| LS-100 | CV-en | 1e-05 | 16 | 0.002 | 54.7 | 22.0 | 61.2 | 26.3 |
| LS-100 | CV-en | 0 | 32 | 0.002 | 54.7 | 20.1 | 61.2 | 23.9 |
| LS-100 | CV-en | 0.1 | 32 | 0.002 | 54.7 | 20.1 | 61.2 | 23.9 |

Table 14: Results (WER, %) on selected FL configurations on CV obtained after $T = 4$k central steps and their comparison to those obtained after $T = 2$k central steps. All parameters are identical to those in Table 4.

| Seed | Data | Cohort size | dev | | | test | | |
|------|------|------|------|------|------|------|------|------|
| | | | $T=0$ | $T=2$k | $T=4$k | $T=0$ | $T=2$k | $T=4$k |
| None | CV-en | 16 | 100.0 | 51.9 | 43.3 | 100.0 | 56.5 | 48.3 |
| None | CV-en | 32 | 100.0 | 41.3 | 34.0 | 100.0 | 46.3 | 38.9 |
| None | CV-en | 64 | 100.0 | 32.9 | 27.3 | 100.0 | 38.0 | 32.0 |
| CV-en-10 | CV-en-90 | 16 | 23.0 | 18.9 | 17.8 | 27.9 | 22.8 | 21.4 |
| CV-en-10 | CV-en-90 | 32 | 23.0 | 17.7 | 16.9 | 27.9 | 21.5 | 20.4 |
| CV-en-10 | CV-en-90 | 64 | 23.0 | 16.7 | 16.0 | 27.9 | 20.1 | 19.4 |
| LS-100 | CV-en | 16 | 54.7 | 22.2 | 19.9 | 61.2 | 26.3 | 23.7 |
| LS-100 | CV-en | 32 | 54.7 | 20.1 | 18.2 | 61.2 | 23.9 | 21.8 |
| LS-100 | CV-en | 64 | 54.7 | 18.4 | 16.8 | 61.2 | 22.0 | 20.2 |

2. Noise is added on the server level after averaging clients' deltas: $\boldsymbol{\Delta}^t = \left[ \frac{1}{L} \sum_{n=1}^{L} \boldsymbol{\Delta}_n^t \right] + \mathcal{N}(0, I\sigma_{avg}^2)$. This is the definition used by McMahan et al. (2018).

3. Noise is added on the server level after summation but before normalization to the number of clients: $\boldsymbol{\Delta}^t = \frac{1}{L} \left[ \left( \sum_{n=1}^{L} \boldsymbol{\Delta}_n^t \right) + \mathcal{N}(0, I\sigma_{sum}^2) \right]$. This definition was used by Abadi et al. (2016).

Different variants of noise are connected with each other via $\sigma_{sum} = \sigma_{avg} \cdot L$ and $\sigma_{client} = \sigma_{avg} \cdot \sqrt{L}$. Then we can compute that $\sigma = \sigma_{avg}$ in this notation from Algorithm 1.

Table 15: Impact of under-trained seed models on WER of the final model for CV dataset with *LS-100* seed and cohort size of 32. The under-trained seed models are obtained from the first 70 steps of the baseline central training used to generate the actual seed model. The parameters for the experiments without seed models and the one with the high quality seed model are the same as in Table 4. The parameters for the seeds of lower quality are the same as those without a seed model.

| Seed | dev | | test | |
|---|---|---|---|---|
| | $T = 0$ | $T = 2k$ | $T = 0$ | $T = 2k$ |
| None | 100.0 | 39.9 | 100.0 | 44.7 |
| LS-100 (30 steps) | 98.9 | 37.7 | 100.0 | 42.8 |
| LS-100 (50 steps) | 83.2 | 32.8 | 87.8 | 37.8 |
| LS-100 (70 steps) | 75.9 | 33.3 | 81.1 | 38.2 |
| LS-100 (full) | 54.7 | 20.1 | 61.2 | 23.9 |

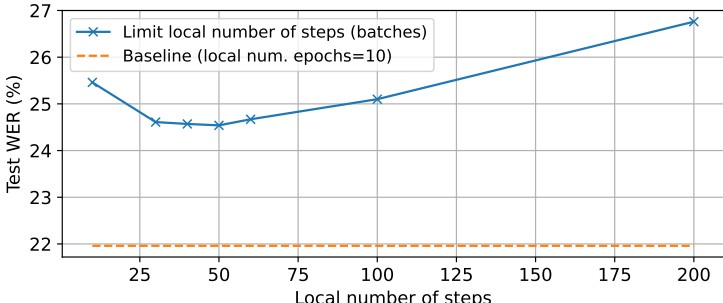

Figure 7: Comparison of WER for FL training between local number of steps (solid) and local number of epochs (dashed). Training is done on *CV-en-train* with a seed model pre-trained centrally on *LS-100*. The cohort size is 64, total number of central steps is $T = 2k$, and all other parameters are set the same as in the corresponding configuration in Figure 3.

Throughout the paper, we use the moments accountant implementation from `opacus` (Yousefpour et al., 2021) which works with $\sigma_{sum}$ noise definition. Thus, to re-scale noise added to each client in order to be consistent with `opacus`, we re-scale it by multiplying by the cohort size $L$ and also by dividing it by the clipping bound $C$. Thus, we get Theorem 1 where the $z$ is defined as $z = \sigma_{sum}$.

In all experiments with FL with DP, we use the same privacy budget for every training step.

### D.2 LARGE COHORT TRAINING IMPLEMENTATION

Our initial FL implementation processed the clients in each cohort sequentially, potentially parallelizing the training for each client using multiple GPUs. For each client, we train a local model for a given number of epochs. However, this approach does not scale well to training with large cohorts, e.g. 1,024, which were necessary for experiments with FL with DP.

That is why we implemented another version where every client is trained on 1 GPU and we train the models for several clients in parallel utilizing all available GPUs (e.g. with 32 GPUs we can process 32 clients in parallel). To do that efficiently with highly imbalanced data like CV where some clients have much more data than others, we restrict the training on every client to a pre-defined number of training steps (batches processed) instead of epochs. Switching from a fixed number of epochs to a fixed number of training steps per client was previously reported to improve performance in the presence of data heterogeneity (Wang et al., 2020).

Since we always use dynamic batching for efficient implementation and the average number of minutes of audio per client in CV is 2.5, FL training with 10 local epochs and total dynamic batch of 2 minutes per client can be approximated with 10 local steps and the same batch size. This configuration is used in all FL with DP experiments.

Unlike reported by Wang et al. (2020), we did not observe improved performance after switching to the number of local steps but instead observed degradation in performance: see Figure 7 for the

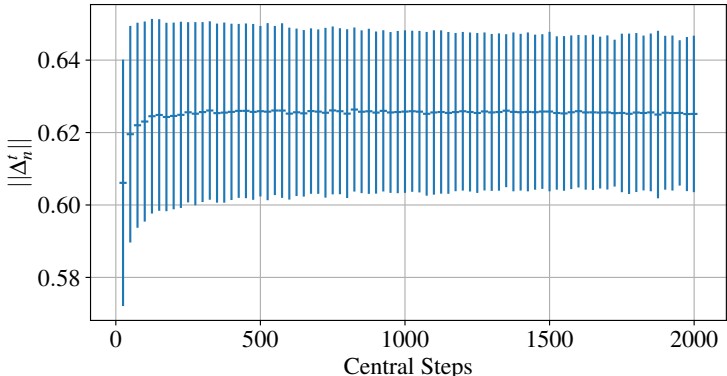

Figure 8: Client's delta norm averaged per clients throughout FL training with the cohort size of $L = 1,024$ on *CV-en-train* from a seed model trained on *LS-100*. We use exponential decay for central LR starting at $t = 750$, decay rate $0.6$, and transition steps $750$ with $T = 2$k total central steps and $10$ local steps. Local (central) LR is $0.2$ ($0.002$).

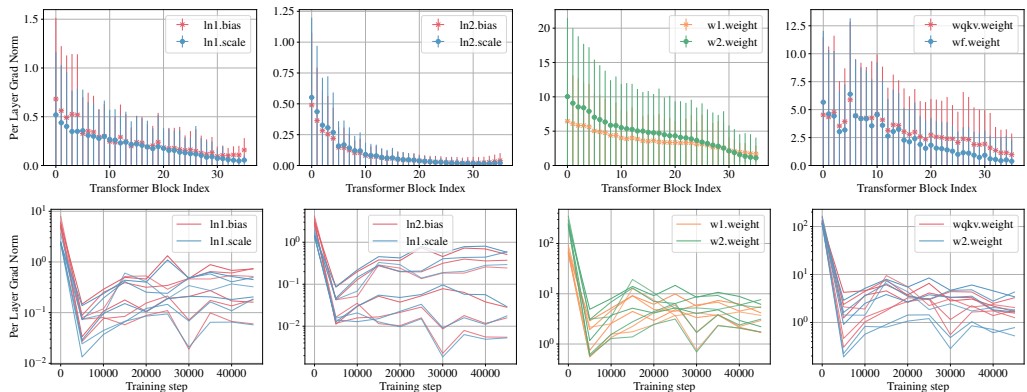

Figure 9: Central training from scratch on *CV-en-train* and its per layer gradients norm: (top) averaged across training steps and (bottom) showed per layer along the training. The model is trained with LARS optimizer and the learning rate of $0.5$. The norms of the per-layer gradients are balanced differently compared to models trained with FL or with FL and DP in Figure 11, e.g., LayerNorm gradients do not dominate over MLP and attention gradients.

results on one configuration of CV with *LS-100* seed model. However, it is of note that the differences will likely get smaller with larger cohort sizes.

## D.3 EMPIRICAL ANALYSIS

For FL training with the large cohort size of $1,024$, the client delta norms are already bounded due to the local clipping (see Algorithm 1, line 10) done in each step of the local training for every client (see Figure 8). Local clipping is a necessary part of the training because otherwise the local training of the transformer model would not converge (Zhai et al., 2023; Dehghani et al., 2023). This is similar to the standard recipe for the central training of transformer models.

As discussed in Section 5.3, we varied the clipping bound $C$ for clients' deltas and did not observe any impact of it on the final performance even when $C = 10^{-8}$. We also did not observe the difference between training with the full precision (float32) or training with the reduced precision (bloat16).

We assume that the seed model is trained centrally without DP[5] (e.g. *LS-100*) after which FL with DP is run on *CV-en-train* by initializing FL model with the seed model. When we add DP noise to the

---

[5]We presume that these data are either public or do not require privacy protection.

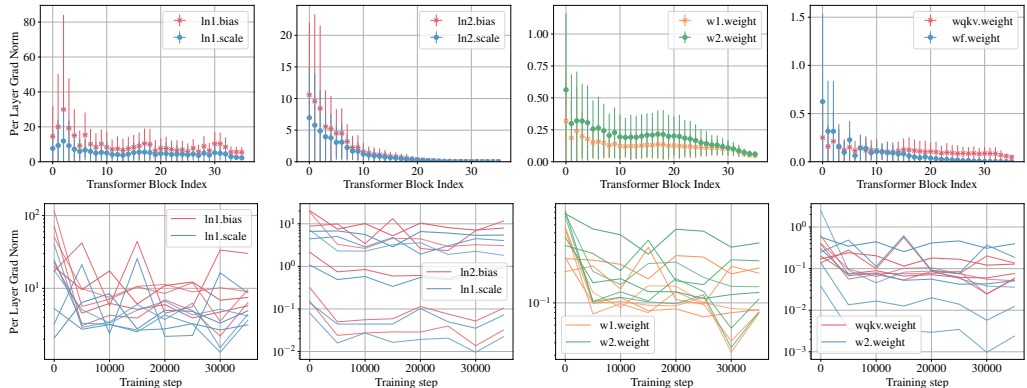

Figure 10: Central training on *CV-en-train* from the *LS-100* seed model and its per layer gradients norm: (top) averaged across training steps and (bottom) showed per layer along the training. The model is trained with LARS optimizer and the learning rate of 0.5. The norms of the per-layer gradients are balanced similarly to models trained with FL or with FL and DP in Figure 11: LayerNorm gradients do dominate over MLP and attention gradients.

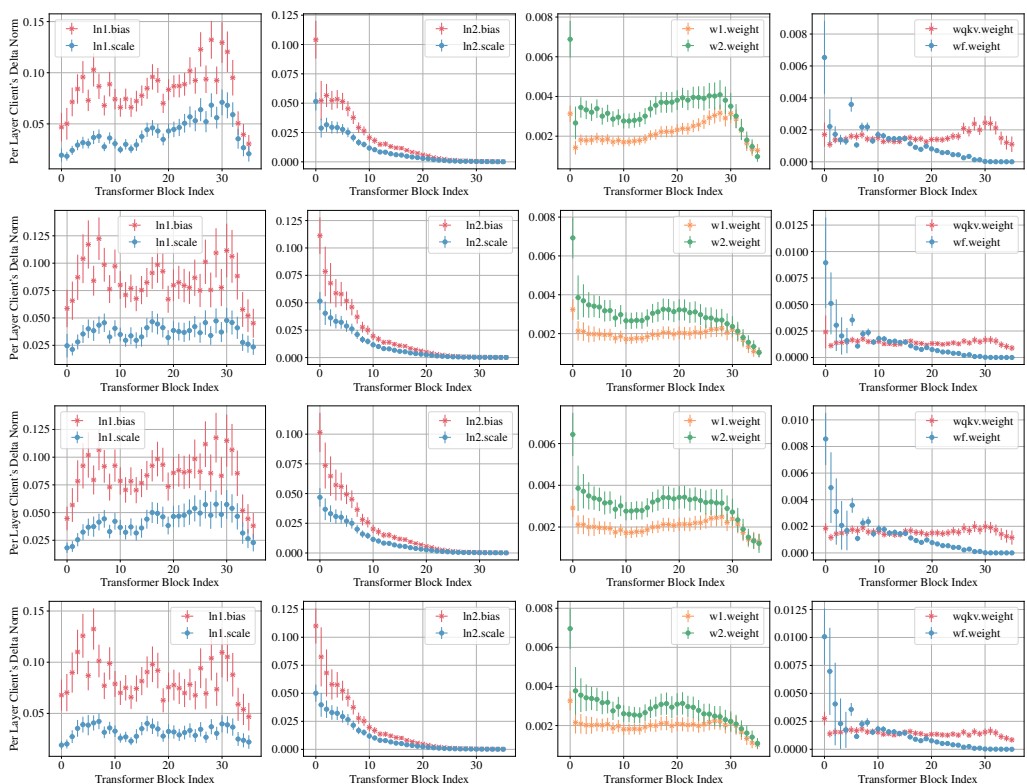

Figure 11: Client delta norms computed per layer in the model. We average the statistics across all clients and central steps, and plot the mean and standard deviation. The model is trained with (first row) global clients' deltas clipping $C = 10^{-2}$ and $\sigma = 0$, (second row) global clients' deltas clipping $C = 10^{-6}$ and $\sigma = 0$, (third row) per-layer clients' deltas clipping (Definition 3, "uniform") $C = 10^{-3}$ and $\sigma = 0$, and (fourth row) per-layer clients' deltas clipping (Definition 3, "uniform") $C = 10^{-2}$ and $\sigma = 3 \cdot 10^{-8}$. The rest of the training configuration is the same as in Figure 5. A transformer block consists of attention parameters (wqkv and wf), MLP (w1 and w2), LayerNorm applied to input of attention (ln1) or MLP (ln2). The statistics are consistent with the training with global clipping (Algorithm 1) in Figure 5.

training alongside with the clipping of clients' deltas, we also did not observe any difference in the training dynamic and final performance (WER) as long as $C/\sigma$ remained constant (e.g., halving the clipping bound $C$ and halving the noise $\sigma$ would produce a nearly identical model). We hypothesise that this is the outcome of (i) above observation that clipping does not affect training; and (ii) using LAMB as a central optimizer, which performs LARS per-layer scaling, and scales both the noise as well as the signal in the same way.

As discussed in Section 5.3, we observe clients' deltas imbalance across different layers of the transformer model (see Figure 5). It is interesting that the first layers (1-10 transformer blocks) have higher delta norms than the last layers (20-36 transformer blocks) for LayerNorm in MLP part and attention final linear projection. This is the opposite behaviour than observed in the deep models, e.g. by Liu et al. (2020). Also, LayerNorms in general have an order of magnitude larger clients' deltas norms than those for MLP and attention. We checked if FL is the source of this deltas imbalance by looking into central training. Interestingly, that central training from scratch on *CV-en-train*, Figure 9, has per layer gradients that behave differently from the clients' deltas in FL or FL with DP training. However, when we compare central training on *CV-en-train* from the same *LS-100* seed model, we will see that per layer gradients behave similarly to the clients' deltas in FL or FL with DP training (see Figure 10).

The smallest delta norms are still non-zero and are order of $10^{-4}$ for LayerNorm (ln2) and $10^{-6}$ for attention (wf) which are re-scaled later by LAMB central optimizer to have the same gradient magnitude across layers. This also highlights necessity of using adaptive optimizers on the server side because otherwise a part of the network will not be trained at all. A similar behavior to the one from Figure 5 can be observed (i) with or without DP noise; and (ii) with global clipping or per-layer clipping of clients' deltas (see Figure 11).

### D.4 DETAILED RESULTS

Comparison for both loss and word error rate (WER) for different values of DP noise and global vs "uniform" per-layer clipping is given in Figure 12, and comparison between "uniform" and "dim" per-layer clipping is given in Figure 13. Training dynamic is shown in Figure 14 for global clipping and in Figure 15 for per-layer clipping. For the per-layer clipping setting we can increase DP noise till $\sigma = 100 \cdot 10^{-8}$ and get similar performance as with global clipping but DP noise $\sigma = 3 \cdot 10^{-8}$. The former is preferable as it has better $(\epsilon, \delta)$-DP guarantees, detailed results of which are shown in Table 16.

### D.5 PER-LAYER CLIPPING ANALYSIS

To understand which part of the transformer is most affected by DP noise, we train a model by adding DP noise only to a particular group of parameters for both global clipping and per-layer "uniform" clipping (see Figure 16): in this case DP guarantees *do not hold*, however we do this for the sake of analysis. We can see that adding DP noise to the parameters of MLP layers drastically reduces model performance, while adding it to other parameters changes WER of the model only marginally. This holds for both types of clipping we apply on clients' deltas.

As per-layer clipping "dim" performed the best in our experiments (see Table 2), we analyse the effect of DP noise for this configuration in depth in Figure 17. First, the results are consistent with Figure 16 in that MLP layers are the most susceptible parts of the transformer, e.g. even if we add DP noise to all layers except MLP ones, we see only small degradation in model performance (middle plot in Figure 16). Second, if we add DP noise with $\sigma$ to all layers but MLP layers get DP noise with $\sigma/2$, we see a significant improvement in the model performance (right plot in Figure 16). The latter suggests that we could redistribute the clipping budget across layers to further alleviate the effect of DP noise during training.

Further experiments with per-layer clipping as $C_i = C\sqrt{\frac{\alpha_i D_i}{\sum_{j=1}^{K} \alpha_j D_j}}$ where $D_i$ is the dimension of the $i$-th layer, $i = \overline{1, K}$, and $\alpha_i = 1$ for all layers except MLP and $\alpha_i = \beta$ for all MLP layers with $\beta \in \{1.5, 2, 3, 10\}$ did not improve results.

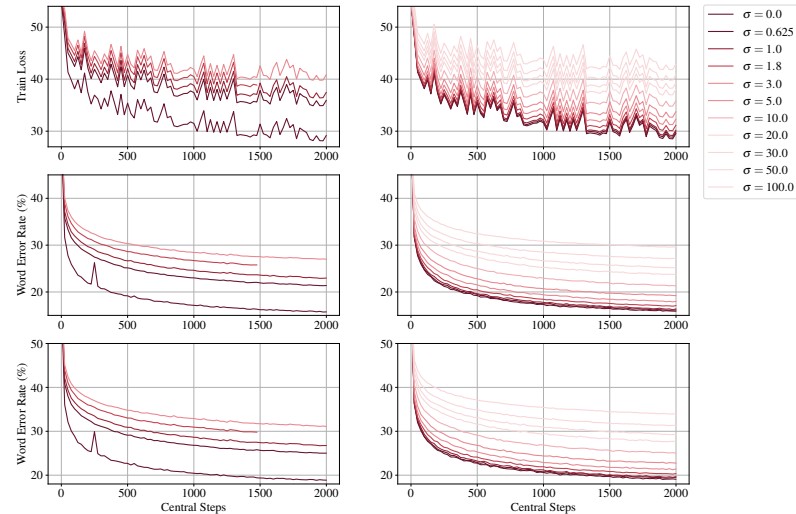

Figure 12: Loss (top) and word error rate (WER) measured on *CV-en-dev* (middle) and *CV-en-test* (bottom) sets for different values of DP noise $\sigma$ (scale is set to $10^{-8}$). We apply clipping of $10^{-2}$ either globally (left, Algorithm 1) or per-layer (right, Definition 3, "uniform") with $T = 2$k central steps and $L =$1,024 cohort size. The rest of the training configuration is the same as in Figure 8. The seed model is trained on *LS-100*.

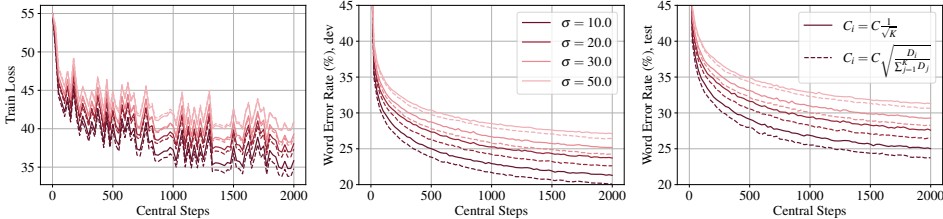

Figure 13: Loss (left) and word error rate (WER) measured on *CV-en-dev* (middle) and *CV-en-test* (right) sets for different values of DP noise $\sigma$ (scale is set to $10^{-8}$). We apply clipping of $10^{-2}$ per-layer (Definition 3, "uniform" and "dim") with $T = 2$k central steps and $L =$1,024 cohort size. The rest of the training configuration is the same as in Figure 8. The seed model is trained on *LS-100*.

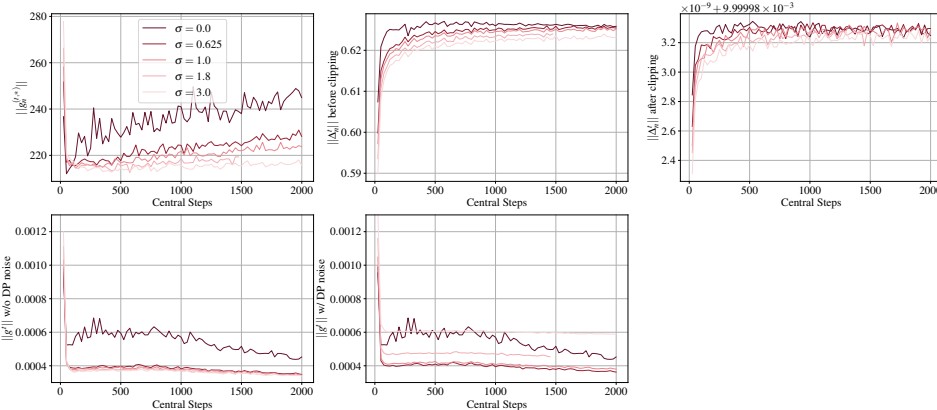

Figure 14: Training dynamic of models from Figure 12 with different DP noise $\sigma$ (scale is set to $10^{-8}$), global clipping of $10^{-2}$ and $T = 2$k central steps. The seed model is trained on *LS-100*: (top, left) client gradients norm during local training (averaged across clients in the cohort); (top, middle) client's delta norm before clipping; (top, right) client's delta norm after clipping; (bottom, left) server gradients norm before DP noise is added per clients' deltas; (bottom, middle) server gradients norm after DP noise is added per clients' deltas.

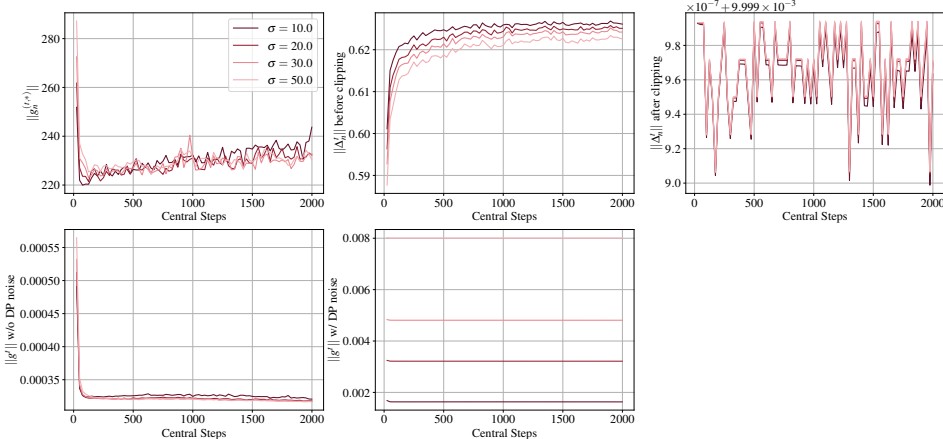

Figure 15: Training dynamic of models from Figure 13 with different DP noise $\sigma$ (scale is set to $10^{-8}$), per-layer clipping of $10^{-2}$ (Definition 3, "dim") and $T = 2k$ central steps. The seed model is trained on *LS-100*: (top, left) client gradients norm during local training (averaged across clients in the cohort); (top, middle) client's delta norm before clipping; (top, right) client's delta norm after clipping; (bottom, left) server gradients norm before DP noise is added per clients' deltas; (bottom, middle) server gradients norm after DP noise is added per clients' deltas.

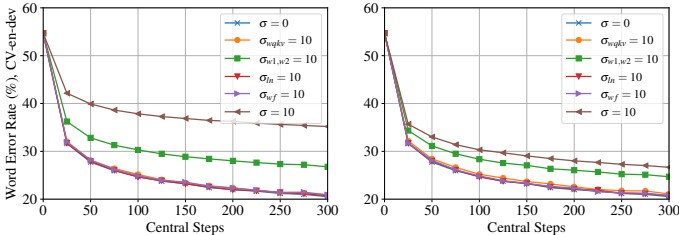

Figure 16: WER of models trained on *CV-en-train* and evaluated on *CV-en-dev* for different values of DP noise $\sigma$ (scale is set to $10^{-8}$). We add either DP noise to all parameters in the model ($\sigma = 10$), or no DP noise ($\sigma = 0$), or DP noise to the specific group of parameters: to attention ($\sigma_{wqkv} = 10$), to MLP ($\sigma_{w1,w2} = 10$), to LayerNorms ($\sigma_{ln} = 10$), to attention final projection ($\sigma_{wf} = 10$). We apply clipping of $10^{-2}$ either globally (left, Algorithm 1) or per-layer (right, Definition 3, "uniform") with $T = 2k$ central steps and $L = 1,024$ cohort size. The rest of the training configuration is the same as in Figure 8. The seed model is trained on *LS-100*.

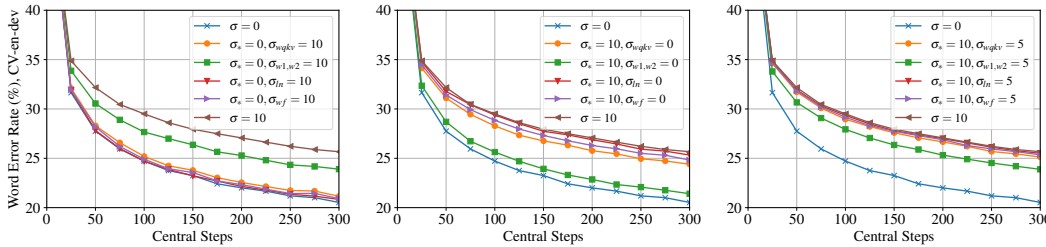

Figure 17: WER of models trained on *CV-en-train* and evaluated on *CV-en-dev* for different values of DP noise $\sigma$ (scale is set to $10^{-8}$). We apply per-layer clipping of $10^{-2}$ (Definition 3, "dim") with $T = 2k$ central steps and $L = 1,024$ cohort size. The rest of the training configuration is the same as in Figure 8. The seed model is trained on *LS-100*. We add either DP noise to all parameters in the model ($\sigma = 10$), or no DP noise ($\sigma = 0$). We also add DP noise (left) to the specific group of parameters only; (middle) to all parameters except the specific group of parameters; (right) to all parameters but the DP noise with $\sigma/2 = 5$ to the specific group of parameters.

Table 16: Extended results of Table 2 for FL with DP and a model pre-trained on *LS-100* ($\sim$100h) used as central data and afterwards fine-tuned on *CV-en-train* ($\sim$1.6k hours) used as clients data. We report added noise $\mathcal{N}(0, I\sigma^2 Nq)$ per client and CV dev and test WERs (%) for two clipping variants with clipping bound $C$: global and per layer "uniform" ("dim"). Total number of users $N$, expected number of users sampled per central step $L = qN$, and the number of central steps $T$ are given. We set $\delta = 10^{-9}$ and report $\epsilon$ for which $(\epsilon, \delta)$-DP holds for a given $L$ and $N$ using the moments accountant of Abadi et al. (2016). For scaling $L$ and $N$ where it is practically intractable to run model training (marked "-"), we extrapolate $(\epsilon, \delta)$-DP assuming training dynamic remains unchanged thus similar WER will be obtained. Central training gives 14.7%/17.8% WER on dev/test. $\epsilon$ should be below 10 to be practically useful (marked with blue).

| $z$ | $\sigma \cdot 10^{-8}$ | $C$ | $L$ | $N$ | $q = L/N$ | $T$ | $\epsilon$ | order | global clipping | | per-layer clipping | |
|---|---|---|---|---|---|---|---|---|---|---|---|---|
| | | | | | | | | | dev WER (%) | test WER (%) | dev WER (%) | test WER (%) |
| - | - | - | 0 | 34,753 | 0 | 0 | 0 | - | 54.7 | 61.2 | 54.7 | 61.2 |
| 0.1024 | 100.0 | 0.01 | 1,024 | 34,753 | 0.0295 | 2,006 | $3.3 \cdot 10^4$ | 1.1 | - | - | 29.6 | 33.9 |
| 1.024 | 100.0 | 0.01 | 10,240 | 347,530 | 0.0295 | 2,006 | $1.3 \cdot 10^1$ | 4.0 | - | - | - | - |
| 5.12 | 100.0 | 0.01 | 51,200 | 1,737,650 | 0.0295 | 2,006 | $1.6 \cdot 10^0$ | 25 | - | - | - | - |
| 0.0512 | 50.0 | 0.01 | 1,024 | 34,753 | 0.0295 | 2,006 | $3.5 \cdot 10^5$ | 1.1 | - | - | 27.1 (26.4) | 31.3 (30.6) |
| 0.512 | 50.0 | 0.01 | 10,240 | 347,530 | 0.0295 | 2,006 | $7.2 \cdot 10^1$ | 1.5 | - | - | - | - |
| 2.56 | 50.0 | 0.01 | 51,200 | 1,737,650 | 0.0295 | 2,006 | $3.5 \cdot 10^0$ | 10.0 | - | - | - | - |
| 0.03072 | 30.0 | 0.01 | 1,024 | 34,753 | 0.0295 | 2,006 | $1.1 \cdot 10^6$ | 1.1 | - | - | 25.2 (24.2) | 29.3 (28.2) |
| 0.3072 | 30.0 | 0.01 | 10,240 | 347,530 | 0.0295 | 2,006 | $3.7 \cdot 10^2$ | 1.1 | - | - | - | - |
| 1.536 | 30.0 | 0.01 | 51,200 | 1,737,650 | 0.0295 | 2,006 | $6.5 \cdot 10^0$ | 7.0 | - | - | - | - |
| 0.02048 | 20.0 | 0.01 | 1,024 | 34,753 | 0.0295 | 2,006 | $2.6 \cdot 10^6$ | 1.1 | - | - | 23.7 (22.6) | 27.6 (26.5) |
| 1.024 | 20.0 | 0.01 | 51,200 | 1,737,650 | 0.0295 | 2,006 | $1.3 \cdot 10^0$ | 4.0 | - | - | - | - |
| 2.048 | 20.0 | 0.01 | 102,400 | 3,475,300 | 0.0295 | 2,006 | $4.5 \cdot 10^0$ | 9.0 | - | - | - | - |
| 0.01024 | 10.0 | 0.01 | 1,024 | 34,753 | 0.0295 | 2,006 | $1.1 \cdot 10^7$ | 1.1 | - | - | 21.3 (20.1) | 25.0 (23.7) |
| 0.512 | 10.0 | 0.01 | 51,200 | 1,737,650 | 0.0295 | 2,006 | $7.2 \cdot 10^1$ | 1.5 | - | - | - | - |
| 0.512 | 10.0 | 0.01 | 51,200 | 17,376,500 | 0.00295 | 2,034 | $1.3 \cdot 10^1$ | 3.0 | - | - | - | - |
| 1.024 | 10.0 | 0.01 | 102,400 | 3,475,300 | 0.0295 | 2,006 | $1.3 \cdot 10^1$ | 4.0 | - | - | - | - |
| 2.048 | 10.0 | 0.01 | 204,800 | 6,950,600 | 0.0295 | 2,006 | $4.5 \cdot 10^0$ | 9.0 | - | - | - | - |
| 2.048 | 10.0 | 0.01 | 204,800 | 69,506,000 | 0.00295 | 2,006 | $7.5 \cdot 10^{-1}$ | 25.0 | - | - | - | - |
| 0.00512 | 5.0 | 0.01 | 1,024 | 34,753 | 0.0295 | 2,006 | $4.2 \cdot 10^7$ | 1.1 | - | - | 19.2 | 22.7 |
| 0.512 | 5.0 | 0.01 | 102,400 | 3,475,300 | 0.0295 | 2,006 | $7.2 \cdot 10^1$ | 1.5 | - | - | - | - |
| 1.024 | 5.0 | 0.01 | 204,800 | 6,950,600 | 0.0295 | 2,006 | $1.3 \cdot 10^1$ | 4.0 | - | - | - | - |
| 1.024 | 5.0 | 0.01 | 204,800 | 69,506,000 | 0.00295 | 2,034 | $2.1 \cdot 10^0$ | 10.0 | - | - | - | - |
| 1.024 | 5.0 | 0.01 | 204,800 | 695,060,000 | 0.000295 | 3,390 | $1.2 \cdot 10^0$ | 15.0 | - | - | - | - |
| 0.003072 | 3.0 | 0.01 | 1,024 | 34,753 | 0.0295 | 2,006 | $1.2 \cdot 10^8$ | 1.1 | 27.0 | 31.1 | 17.9 (17.1) | 21.2 (20.4) |
| 0.3072 | 3.0 | 0.01 | 102,400 | 3,475,300 | 0.0295 | 2,006 | $3.7 \cdot 10^2$ | 1.1 | - | - | - | - |
| 0.6144 | 3.0 | 0.01 | 204,800 | 6,950,600 | 0.0295 | 2,006 | $4.2 \cdot 10^1$ | 2.0 | - | - | - | - |
| 0.6144 | 3.0 | 0.01 | 204,800 | 69,506,000 | 0.00295 | 2,034 | $7.2 \cdot 10^0$ | 3.0 | - | - | - | - |
| 0.6144 | 3.0 | 0.01 | 204,800 | 695,060,000 | 0.000295 | 3,390 | $3.7 \cdot 10^0$ | 6.0 | - | - | - | - |
| 0.0018432 | 1.8 | 0.01 | 1,024 | 34,753 | 0.0295 | 2,006 | $4.5 \cdot 10^8$ | 1.5 | 25.8 | 29.2 | 17.0 | 20.2 |
| 0.18432 | 1.8 | 0.01 | 102,400 | 3,475,300 | 0.0295 | 2,006 | $2.3 \cdot 10^4$ | 1.5 | - | - | - | - |
| 0.36864 | 1.8 | 0.01 | 204,800 | 6,950,600 | 0.0295 | 2,006 | $2.7 \cdot 10^2$ | 1.5 | - | - | - | - |
| 0.36864 | 1.8 | 0.01 | 204,800 | 69,506,000 | 0.00295 | 2,034 | $4.5 \cdot 10^1$ | 1.5 | - | - | - | - |
| 0.36864 | 1.8 | 0.01 | 204,800 | 695,060,000 | 0.000295 | 3,390 | $1.6 \cdot 10^1$ | 2.5 | - | - | - | - |
| 0.001024 | 1.0 | 0.01 | 1,024 | 34,753 | 0.0295 | 2,006 | $1.1 \cdot 10^9$ | 1.1 | 22.9 | 26.7 | 16.2 (16.0) | 19.5 (19.3) |
| 0.1024 | 1.0 | 0.01 | 102,400 | 3,475,300 | 0.0295 | 2,006 | $3.2 \cdot 10^4$ | 1.1 | - | - | - | - |
| 0.2048 | 1.0 | 0.01 | 204,800 | 6,950,600 | 0.0295 | 2,006 | $1.1 \cdot 10^3$ | 1.1 | - | - | - | - |
| 0.2048 | 1.0 | 0.01 | 204,800 | 69,506,000 | 0.00295 | 2,034 | $2.7 \cdot 10^2$ | 1.1 | - | - | - | - |
| 0.2048 | 1.0 | 0.01 | 204,800 | 695,060,000 | 0.000295 | 3,390 | $9.4 \cdot 10^1$ | 1.3 | - | - | - | - |
| 0.0006144 | 0.625 | 0.01 | 1,024 | 34,753 | 0.0295 | 2,006 | $4.0 \cdot 10^9$ | 1.5 | 21.3 | 25.0 | 16.1 | 19.3 |
| 0.06144 | 0.625 | 0.01 | 102,400 | 3,475,300 | 0.0295 | 2,006 | $3.8 \cdot 10^5$ | 1.5 | - | - | - | - |
| 0.12288 | 0.625 | 0.01 | 204,800 | 6,950,600 | 0.0295 | 2,006 | $7.9 \cdot 10^4$ | 1.5 | - | - | - | - |
| - | 0 | 0.01 | 1,024 | 34,753 | 0.0295 | 2,000 | inf | - | 15.7 | 18.9 | 15.9 | 19.1 |
| - | 0 | 1.0 | 1,024 | 34,753 | 0.0295 | 2,000 | inf | - | 15.7 | 18.9 | - | - |

Table 17: Results for FL with DP and a model pre-trained on *CV-fr-train-10/CV-de-train-10* (∼50h) used as central data and afterwards fine-tuned on (top/bottom) *CV-fr-train-90/CV-de-train-90* (∼700-800 hours) used as clients data. We report added noise $\mathcal{N}(0, I\sigma^2 Nq)$ per client and CV dev and test WERs (%) for two clipping variants with clipping bound $C$: global and per layer "dim". Total number of users $N$, expected number of users sampled per central step $L = qN$, and the number of central steps $T$ are given. We set $\delta = 10^{-9}$ and report $\epsilon$ for which $(\epsilon, \delta)$-DP holds for a given $L$ and $N$ using the moments accountant of Abadi et al. (2016). For scaling $L$ and $N$ where it is practically intractable to run model training (marked "-"), we extrapolate $(\epsilon, \delta)$-DP assuming training dynamic remains unchanged thus similar WER will be obtained. Central training gives 10.8%/12.6% WER for French and 8.1%/9.2% WER for German on dev/test. $\epsilon$ should be below 10 to be practically useful (marked with blue).

| $z$ | $\sigma \cdot 10^{-8}$ | $C$ | $L$ | $N$ | $q = L/N$ | $T$ | $\epsilon$ | order | global clipping | | per-layer clipping "dim" | |
|---|---|---|---|---|---|---|---|---|---|---|---|---|
| | | | | | | | | | dev WER (%) | test WER (%) | dev WER (%) | test WER (%) |
| - | - | - | 0 | 6,171 | 0 | 0 | 0 | - | 24.0 | 27.5 | 24.0 | 27.5 |
| 0.01024 | 10.0 | 0.01 | 1,024 | 6,171 | 0.1660 | 2,002 | $1.1 \cdot 10^7$ | 1.3 | - | - | 15.6 | 17.9 |
| 2.56 | 10.0 | 0.01 | 256,000 | 1,542,750 | 0.1660 | 2,002 | $2.4 \cdot 10^1$ | 3.0 | - | - | - | - |
| 2.56 | 10.0 | 0.01 | 256,000 | 15,427,500 | 0.0166 | 2,013 | $1.9 \cdot 10^0$ | 20.0 | - | - | - | - |
| 0.003072 | 3.0 | 0.01 | 1,024 | 6,171 | 0.1660 | 2,002 | $1.2 \cdot 10^8$ | 1.1 | 14.1 | 16.2 | 13.9 | 16.0 |
| 0.768 | 3.0 | 0.01 | 256,000 | 1,542,750 | 0.1660 | 2,002 | $1.8 \cdot 10^2$ | 3.0 | - | - | - | - |
| 0.768 | 3.0 | 0.01 | 256,000 | 15,427,500 | 0.0166 | 2,013 | $1.4 \cdot 10^1$ | 3.0 | - | - | - | - |
| 0.768 | 3.0 | 0.01 | 256,000 | 46,282,500 | 0.00553 | 1,991 | $5.5 \cdot 10^0$ | 5.0 | - | - | - | - |
| - | 0 | 0.01 | 1,024 | 6,171 | 0.1660 | 2,000 | inf | - | 13.2 | 15.2 | 13.2 | 15.2 |
| - | - | - | 0 | 6,415 | 0 | 0 | 0 | - | 18.6 | 21.2 | 18.6 | 21.2 |
| 0.01024 | 10.0 | 0.01 | 1,024 | 6,415 | 0.1596 | 2,002 | $1.1 \cdot 10^7$ | 1.1 | - | - | 12.3 | 13.9 |
| 2.56 | 10.0 | 0.01 | 256,000 | 1,603,750 | 0.1596 | 2,002 | $2.3 \cdot 10^1$ | 3.0 | - | - | - | - |
| 2.56 | 10.0 | 0.01 | 256,000 | 16,037,500 | 0.01596 | 2,016 | $1.8 \cdot 10^0$ | 20.0 | - | - | - | - |
| 0.003072 | 3.0 | 0.01 | 1,024 | 6,415 | 0.1596 | 2,002 | $1.2 \cdot 10^8$ | 1.1 | 10.7 | 12.1 | 10.5 | 12.0 |
| 0.768 | 3.0 | 0.01 | 256,000 | 1,603,750 | 0.1596 | 2,002 | $1.7 \cdot 10^2$ | 1.5 | - | - | - | - |
| 0.768 | 3.0 | 0.01 | 256,000 | 16,037,500 | 0.01596 | 2,016 | $1.4 \cdot 10^1$ | 4.0 | - | - | - | - |
| 0.768 | 3.0 | 0.01 | 256,000 | 48,112,500 | 0.00532 | 2,068 | $5.4 \cdot 10^0$ | 5.0 | - | - | - | - |
| - | 0 | 0.01 | 1,024 | 6,415 | 0.1596 | 2,000 | inf | - | 9.7 | 11.0 | 9.7 | 11.0 |

Table 18: Ablation for FL with DP and a model pre-trained either on *LS-960/CV-en-train-10* used as central data and afterwards fine-tuned on (top/bottom) *CV-en-train/CV-en-train-90*. We report added noise $\mathcal{N}(0, I\sigma^2 Nq)$ per client and CV dev and test WERs (%) for two clipping variants with clipping bound $C$: global and per layer "dim". Total number of users $N$, expected number of users sampled per central step $L = qN$, and the number of central steps $T$ are given. Central training gives 14.1%/17.2% WER for training from *LS-960* seed and 14.5%/17.6% for training from *CV-en-train-10* seed on dev/test. All the remaining parameters are the same as in Table 16.

| Seed | Data | $\sigma \cdot 10^{-8}$ | $C$ | $L$ | $N$ | $q = L/N$ | $T$ | global clipping | | per-layer clipping "dim" | |
|---|---|---|---|---|---|---|---|---|---|---|---|
| | | | | | | | | dev WER (%) | test WER (%) | dev WER (%) | test WER (%) |
| LS-960 | - | - | - | 0 | 34,753 | 0 | 0 | 27.0 | 31.5 | 27.0 | 31.5 |
| LS-960 | CV-en-train | 30 | 0.01 | 256 | 34,753 | 0.0074 | 2000 | 22.5 | 26.1 | 18.7 | 22.2 |
| CV-10 | - | - | - | 0 | 34,753 | 0 | 0 | 23.0 | 27.9 | 23.0 | 27.9 |
| CV-10 | CV-en-train-90 | 30 | 0.01 | 256 | 31,278 | 0.0082 | 2000 | 20.8 | 25.1 | 18.7 | 22.6 |

## D.6 FEDERATED LEARNING WITH DIFFERENTIAL PRIVACY FOR FRENCH AND GERMAN

We run out of the box experiments for FL with DP for French and German CV data using the same configuration as for English (training parameters are given in Figure 8). Seed models are trained on *CV-fr-train-10* and *CV-de-train-10*, while *CV-fr-train-90* and *CV-de-train-90* are used for further FL with DP training. We get similar results as for English, see Table 17. With the same DP noise $\sigma = 3 \cdot 10^{-8}$, we are able to closely match the model trained without DP noise ($\sigma = 0$) with only a small WER degradation: (i) for French from 15.2% to 16.0% WER while guaranteeing $(5.5, 10^{-9})$-DP, and (ii) for German from 11.0% to 12.0% WER while guaranteeing $(5.4, 10^{-9})$-DP; assuming the training effectiveness remains the same if we extrapolate to ∼50M clients with the cohort size of ∼250k. Moreover, we can also increase DP noise to $\sigma = 10^{-7}$, getting 17.9% WER with $(1.9, 10^{-9})$-DP for French and 13.9% WER with $(1.8, 10^{-9})$-DP for German by scaling only to ∼16M clients with the cohort size of ∼250k, assuming the training effectiveness remains the same. The latter is a realistic scenario for mid/low resource languages.

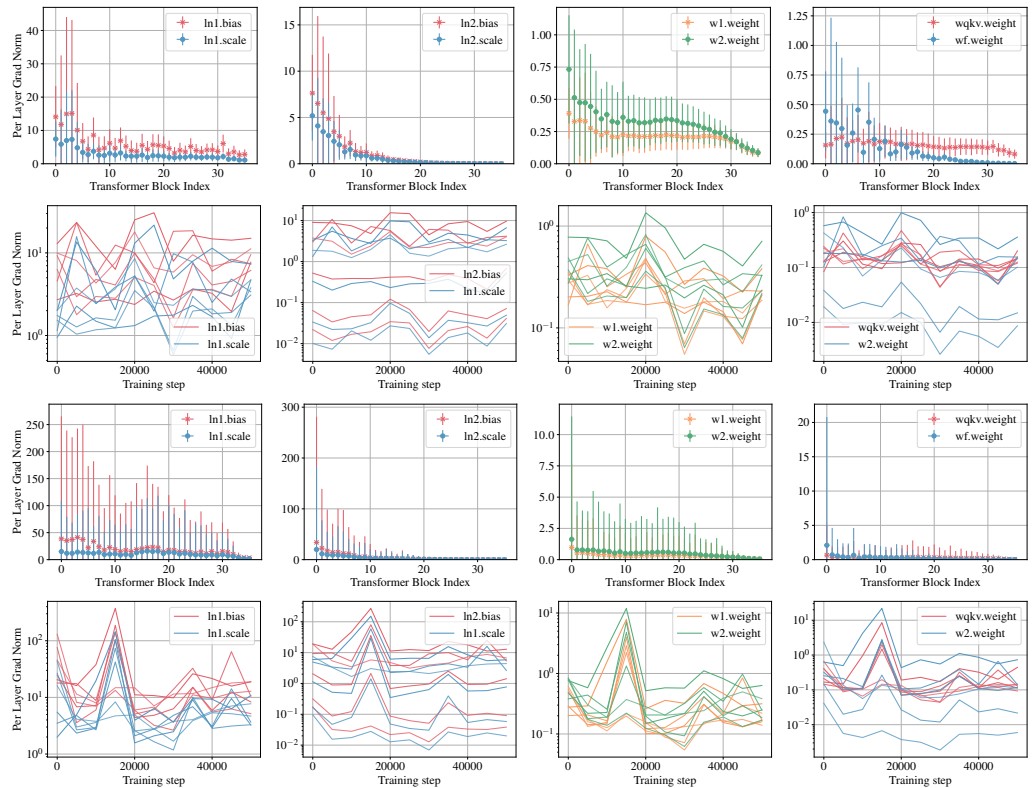

Figure 18: (first and second rows) Central training on *CV-en-train* from the *LS-960* seed model and (third and fourth rows) Central training on *CV-en-train-90* from the *CV-en-train-10* seed model and their per layer gradients norm: (first, third rows) averaged across training steps and (second, fourth) showed per layer along the training. The model is trained with LARS optimizer and the learning rate of 0.5/0.2. LayerNorm gradients **do** dominate over MLP and attention gradients.

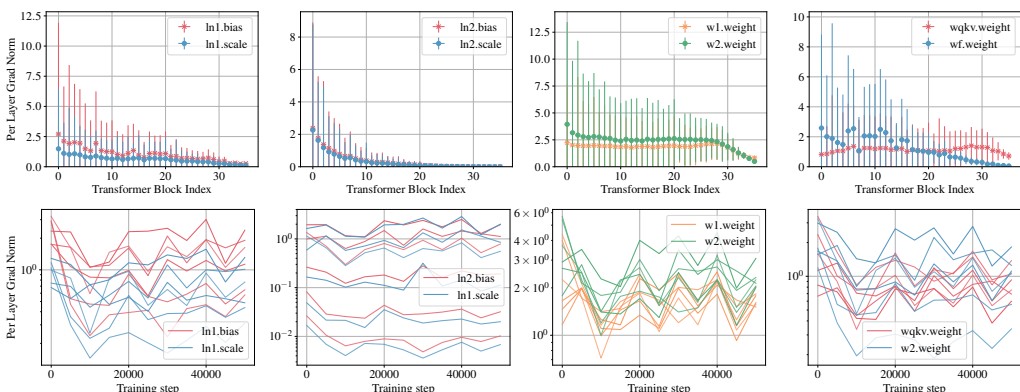

Figure 19: Central training on *CV-fr-train-90* from the *CV-fr-train-10* seed model and its per layer gradients norm: (top) averaged across training steps and (bottom) showed per layer along the training. The model is trained with LARS optimizer and the learning rate of 0.2. The norms of the per-layer gradients are balanced similarly to models trained with FL or with FL and DP in Figure 21: LayerNorm gradients **do not** dominate over MLP and attention gradients.

Interestingly, for both French and German we observe that per-layer clipping is not as effective as for English and we get only marginal improvements over global clipping. We have checked that the seed model quality and the seed model being out-of-domain are the not the sources of this discrepancy in results between languages: if we change the seed model for English to a better out-of-domain

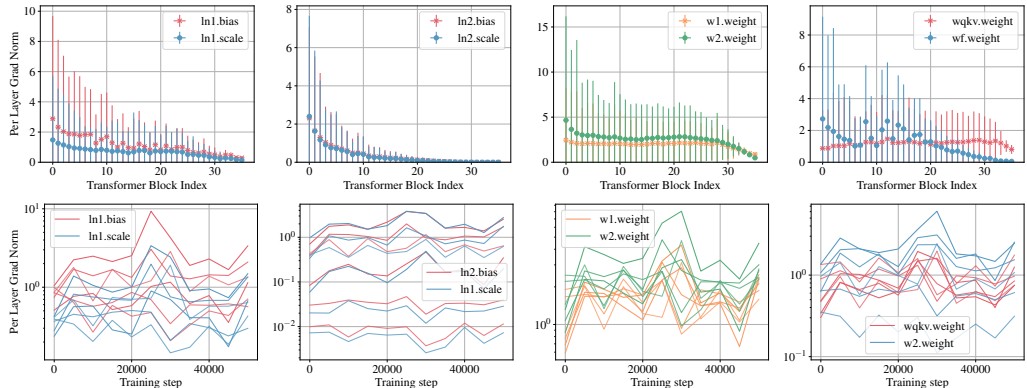

Figure 20: Central training on *CV-de-train-90* from the *CV-de-train-10* seed model and its per layer gradients norm: (top) averaged across training steps and (bottom) showed per layer along the training. The model is trained with LARS optimizer and the learning rate of 0.2. The norms of the per-layer gradients are balanced similarly to models trained with FL or with FL and DP in Figure 22: LayerNorm gradients **do not** dominate over MLP and attention gradients.

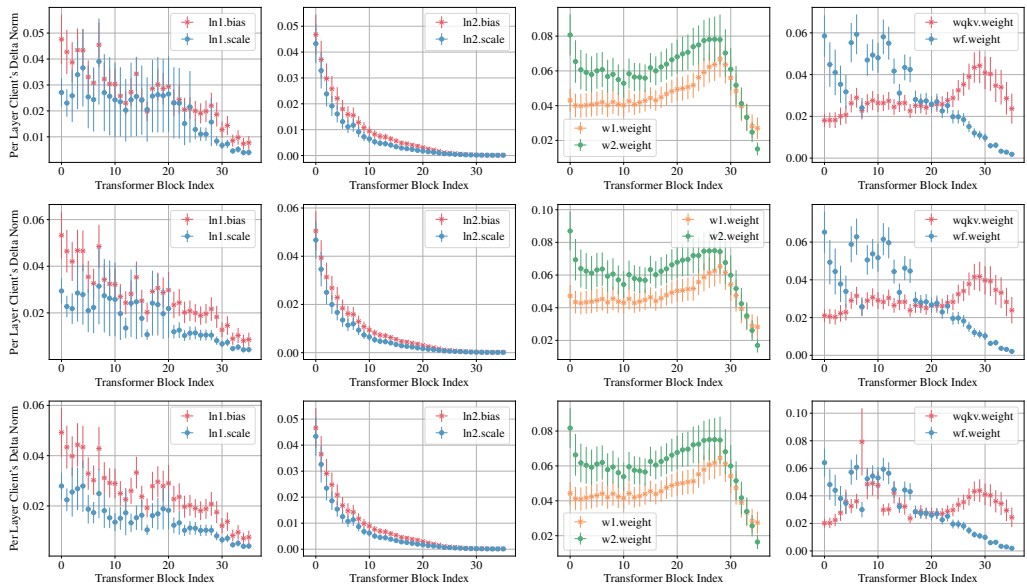

Figure 21: Client delta norms computed per layer in the French model trained on *CV-fr-train-90* from a seed *CV-fr-train-10* model. We average the statistics across all clients and central steps, and plot the mean and standard deviation. The model is trained with (first row) global clients' deltas clipping $C = 10^{-2}$ and $\sigma = 0$, (second row) global clients' deltas clipping $C = 10^{-2}$ and $\sigma = \sigma = 3 \cdot 10^{-8}$, (third row) per-layer clients' deltas clipping (Definition 3, "dim") $C = 10^{-2}$ and $\sigma = 3 \cdot 10^{-8}$. The rest of the training configuration is the same as in Figure 5. A transformer block consists of attention parameters (wqkv and wf), MLP (w1 and w2), LayerNorm applied to input of attention (ln1) or MLP (ln2).

*LS-960* seed or to a better in-domain *CV-en-train-10* seed, we still observe a drastic improvement from per-layer clipping compared to global clipping (see Table 18, and Figure 18).

First, there is a discrepancy in gradients balance across layers for the central model training for English, French and German with *CV-\*-train-10* seed models. The training of the English model has the issue we discussed above that LayerNorms dominate the attention and MLP, which translates to the similar behavior for FL and FL with DP training. However, French and German models do not have the same imbalance issue as English and, moreover, similar behavior holds for the central

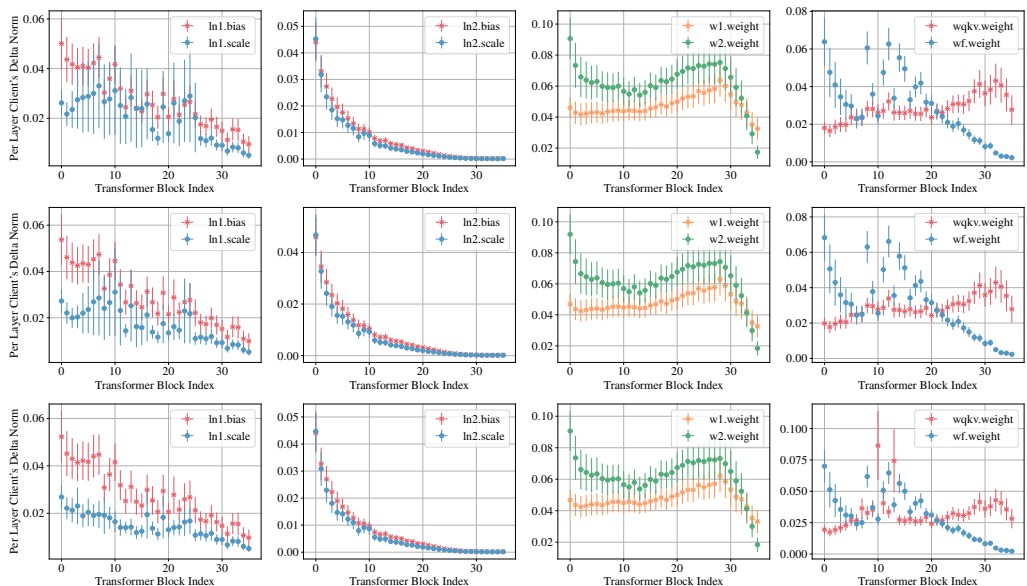

Figure 22: Client delta norms computed per layer in the German model trained on *CV-de-train-90* from a seed *CV-de-train-10* model. We average the statistics across all clients and central steps, and plot the mean and standard deviation. The model is trained with (first row) global clients' deltas clipping $C = 10^{-2}$ and $\sigma = 0$, (second row) global clients' deltas clipping $C = 10^{-2}$ and $\sigma = \sigma = 3 \cdot 10^{-8}$, (third row) per-layer clients' deltas clipping (Definition 3, "dim") $C = 10^{-2}$ and $\sigma = 3 \cdot 10^{-8}$. The rest of the training configuration is the same as in Figure 5. A transformer block consists of attention parameters (wqkv and wf), MLP (w1 and w2), LayerNorm applied to input of attention (ln1) or MLP (ln2).

training, FL and FL with DP for French and German (see Figures 19, 20, 21 and 22). We attribute the later to the properties of the languages as discussed in Appendix C.4.

One factor that we cannot exclude from the above analysis is the user sampling $q = L/N$, which is significantly higher for French and German (16%) than for English ($< 1\%$) due to a smaller number of speakers in the French and German datasets. Further investigation is needed to evaluate larger datasets with a larger number of speakers for French and German (as we need a large cohort size to alleviate the impact of DP noise), and to probe other languages.

