# OpenReview forum: "Federated Learning with Differential Privacy for End-to-End Speech Recognition"
_ICLR.cc/2024/Conference — ICLR 2024 Conference Withdrawn Submission_

### Official Review · Reviewer_Jahb · 2023-10-21

**Soundness:** 2 fair
**Presentation:** 2 fair
**Contribution:** 2 fair
**Rating:** 3
**Confidence:** 3

**Summary:**

This paper trains ASR models with user-level differential privacy in a federated learning setting. The authors use two main techniques to improve the model performance. The first is using a pre-trained model (seed model) as initialization. The second is applying layer-wise clipping when running DP-SGD. The largest dataset in this paper consists of data from ~40K users, which is not enough to derive a meaningful privacy bound. The authors show the privacy guarantee could be acceptable if the number of users is hypothetically scaled to ~6.9M.

**Strengths:**

This is the first paper studying DP training of ASR models in the FL setting. The findings presented in this study offer valuable insights and underscore the challenges in this area.

**Weaknesses:**

1. In Table 2, the authors hypothetically scale the number of users to ~6.9M. This is hard to achieve even in a central setting, and will be even harder considering the implementation challenges in the FL setting. Therefore, it is not clear whether the findings have practical meaning.

2. The seed model in Table 2 is only pre-trained on LS-100, whose size is more than 10 times smaller than the fine-tuning dataset. Why use such a small pre-training dataset? Recent work in this area shows that private learning significantly benefits from better pre-trained models, e.g., see [1] and the references therein. The authors may want to explore pre-training the model on increasingly large datasets, e.g., from LS-100 to LS-960, and see whether better pre-trained models help private training of ASR models.

3. He et al., 2023 [2] also use pre-layer clipping, although their focus is NLP tasks. They also observe the difference in gradient norms of different transformer layers. However, they show that per-layer clipping only achieves comparable or slightly better results compared to global clipping. It would be interesting to the community if the authors could explain why per-layer clipping is much more important in the training of ASR models.


References:

[1]: Why Is Public Pretraining Necessary for Private Model Training? https://arxiv.org/abs/2302.09483

[2]: Exploring the Limits of Differentially Private Deep Learning with Group-wise Clipping. https://arxiv.org/abs/2212.01539

**Questions:**

Please see `Weaknesses`

---

> ### Author Response · Authors · 2023-11-17
> **Authors' response**
>
> We thank Reviewer Jahb for their time and feedback. Please find below our comments:
>
> > In Table 2, the authors hypothetically scale the number of users to ~6.9M.
>
> We would like to scale to 7M speakers (which is a practical number for mid-resource languages), however we are not aware of any publicly available ASR dataset of that scale. In practice, if we have 2.5 min per user (like in CV) and 7M users total, training of our model for central simulation of FL with 1024 GPUs, cohort size of 200k, and 2k central steps would take 10 days (see Table below for the training we have run for CV).
>
> | Seed | Data | Model | Client Total Batch Size | Cohort Size $L$ | Local | Central Steps $T$ | # GPUs A100 40GB | Runtime (h) | Total GPU (h) |
> |--- |--- |--- |--- |--- |--- |--- |--- |--- |--- |
> | LS-100 | CV-en-train | FL | 2min | 1,024 | 10 steps | 2000 | 32 | 34 | 1,088 |
> | LS-100 | CV-en-train | FL + DP | 2min | 1,024 | 10 steps | 2000 | 32 | 35 | 1,120 |
> | LS-100 | CV-en-train | FL + DP | 2min | 256 | 10 steps | 2000 | 16 | 18 | 288 |
> | CV-fr-train-10 | CV-fr-train-90 | FL | 2min | 1,024 | 10 steps | 2000 | 16 | 60 | 960 |
> | CV-fr-train-10 | CV-fr-train-90 | FL + DP | 2min | 1,024 | 10 steps | 2000 | 16 | 61 | 976 |
> | CV-fr-train-10 | CV-fr-train-90 | FL + DP | 2min | 1,024 | 10 steps | 2000 | 64 | 18 | 1,152 |
>
> In fact, training depends on cohort size and central iterations, but not the population size.
>
> > This is hard to achieve even in a central setting, and will be even harder considering the implementation challenges in the FL setting. Therefore, it is not clear whether the findings have practical meaning.
>
> We would like to highlight that prior works, e.g. Mike Chatzidakis et.al. (2023), Learning iconic
> scenes with differential privacy, https://machinelearning.apple.com/research/scenes-differential-privacy, and Xu et al. (2023), Federated Learning of Gboard Language Models with Differential Privacy, showed i) it is **realistic** to have **millions of users** to participate in FL and ii) use a large cohort size of 150k in FL deployments. Indeed, practical deployments of PFL described in those works work at that scale.
>
> We acknowledge that there are additional challenges such as communication cost that need to be addressed for successful deployment of the proposed ASR FL+DP, and believe that further research on the problem will help alleviate this issue. In this paper, we focus on understanding limitations of **large-scale ASR** in FL and FL+DP with e.g. practical i) central number of steps; ii) data distribution per user; iii) out-of-domain shift.

---

> ### Author Response · Authors · 2023-11-17
> **Authors' response [continue]**
>
> > The seed model in Table 2 is only pre-trained on LS-100, whose size is more than 10 times smaller than the fine-tuning dataset. Why use such a small pre-training dataset? Recent work in this area shows that private learning significantly benefits from better pre-trained models, e.g., see [1] and the references therein. The authors may want to explore pre-training the model on increasingly large datasets, e.g., from LS-100 to LS-960, and see whether better pre-trained models help private training of ASR models.
>
> We use LS-100 for seed model training to simulate **practical scenario** where i) central data are an **order of magnitude smaller (16x in our case)** than federated data; ii) central data are of reasonable size for mid-resource languages as labeling process is expensive.
>
> At the same time, we did ablation with LS-960 seed model in Appendix D.5, Table 18. We have run more ablations on DP, see Table below. Overall, results are similar to LS-100 seed model and per-layer clipping improves WER significantly for the same DP guarantees. However, indeed, the DP noise can be increased by 3x ($30\cdot 10^{-8}$ instead of $10\cdot 10^{-8}$) with similar 30% relative WER degradation (17.8% w/o DP drops to 22.2% with DP): then in theory, we can use 50k cohort size and 2M population for practical DP guarantees ($\epsilon=6.5$). Though, it could be due to 1:1 proportion between central and federated data, which is less realistic scenario.
>
> | $z$ | $\sigma\cdot 10^{-8}$ | $C$ | $L$ | $N$ | $q=L/N$ | $T$ | $\epsilon$ | order | [global] dev WER | [global] test WER | [per-layer, dim] dev WER | [per-layer, dim] test WER |
> |--- |--- |--- |--- |--- |--- |--- |--- |--- |--- |--- |--- |--- |
> | - | - | - | 0 | 34,753 | 0 | 0 | 0 | - | 27.0 | 31.5 | 27.0 | 31.5 |
> |0.03072 | $30.0$ | 0.01 | 1,024 | 34,753 | 0.0295 | 2,006 | 1.1$\cdot 10^6$ | 1.1 | 22.5 | 26.1 | 18.7 | 22.2 |
> |0.3072 | $30.0$ | 0.01 | 10,240 | 347,530 | 0.0295 | 2,006 | 3.7$\cdot 10^2$ | 1.1 | - | - | - | - |
> |1.536 | $30.0$ | 0.01 | 51,200 | 1,737,650 | 0.0295 | 2,006 | 6.5$\cdot 10^0$ | 7.0 | - | - | - | - |
> |0.01024 | $10.0$ | 0.01 | 1,024 | 34,753 | 0.0295 | 2,006 | 1.1$\cdot 10^7$ | 1.1 | 20.5 | 24.1 | 16.5 | 19.7 |
> |0.512 | $10.0$ | 0.01 | 51,200 | 1,737,650 | 0.0295 | 2,006 | 7.2$\cdot{10^1}$ | 1.5 | - | - | - | - |
> |0.512 | $10.0$ | 0.01 | 51,200 | 17,376,500 | 0.00295 | 2,034 | 1.3$\cdot 10^1$ | 3.0 | - | - | - | - |
> |1.024 | $10.0$ | 0.01 | 102,400 | 3,475,300 | 0.0295 | 2,006 | 1.3$\cdot 10^1$ | 4.0 | - | - | - | - |
> |2.048 | $10.0$ | 0.01 | 204,800 | 6,950,600 | 0.0295 | 2,006 | 4.5$\cdot 10^0$ | 9.0 | - | - | - | -|
> |2.048 | $10.0$ | 0.01 | 204,800 | 69,506,000 | 0.00295 | 2,006 | 7.5$\cdot 10^{-1}$ | 25.0 | - | - | - | - |
> |0.003072 | $3.0$ | 0.01 | 1,024 | 34,753 | 0.0295 | 2,006 | 1.2$\cdot 10^8$ | 1.1 | 18.1 | 21.6 | 14.9 | 17.8 |
> |0.3072 | $3.0$ | 0.01 | 102,400 | 3,475,300 | 0.0295 | 2,006 | 3.7$\cdot 10^2$ | 1.1 | - | - | - | - |
> |0.6144 | $3.0$ | 0.01 | 204,800 | 6,950,600| 0.0295 | 2,006 | 4.2$\cdot 10^1$ | 2.0 | - | - | - | -|
> |0.6144 | $3.0$ | 0.01 | 204,800 | 69,506,000| 0.00295 | 2,034 | 7.2$\cdot 10^0$ | 3.0 | - | - | - | - |
> |0.6144 | $3.0$ | 0.01 | 204,800 | 695,060,000 | 0.000295 | 3,390 | 3.7$\cdot 10^0$ | 6.0 | - | - | - | - |
> | - | 0 | 0.01 | 1,024 | 34,753 | 0.0295 | 2,000 | $\inf$ | - | 13.9 | 16.7 | 14.0 | 16.8 |
>
> > He et al., 2023 [2] also use pre-layer clipping, although their focus is NLP tasks. They also observe the difference in gradient norms of different transformer layers. However, they show that per-layer clipping only achieves comparable or slightly better results compared to global clipping. It would be interesting to the community if the authors could explain why per-layer clipping is much more important in the training of ASR models.
>
> Thanks for the pointer! Also original work of McMahan et al. (2018) showed that per-layer clipping achieves similar results as global clipping (see discussion in Section 3). For ASR:
> - We study it in detail and observe different behaviour in Appendices D.5 and D.6 where we exclude different confound factors, like speech language, seed models, optimizers, gradient clipping, DP, federated training. We observe that imbalanced gradients occur already in central training and depend on the speech language: they occur for English but not for German and French.
> - We attribute it to the complexity of the language / data, and central training has a similar behavior to FL and FL + DP.
>
> We would like to highlight
> - Our architecture is encoder-based trained with a sequence loss (CTC), while in the pointed work it is decoder-based (causal) with cross-entropy loss for NLP which makes models totally different for optimization.
> - Tables 3, 4 of He et al., 2023 shows that, per-layer clipping significantly improves results for GLUE tasks for some settings.
> - Moreover,  He et al., 2023 fine-tunes pretrained model for the downstream task with another objective, while in ASR we keep it the same.

---

### Official Review · Reviewer_JcFR · 2023-10-23

**Soundness:** 3 good
**Presentation:** 3 good
**Contribution:** 1 poor
**Rating:** 3
**Confidence:** 4

**Summary:**

Primary contribution is the application of FL to a heterogeneous automatic speech recognition task, secondary is the application of DP to this FL setting.

**Strengths:**

This work is very well-written, the idea is justified and the motivation is formulated very clearly. There are a multitude of results in the main body and the appendix, so clearly many hours were put into this work (as well as polishing the manuscript, I could not find any obvious typos etc.)

The results are very promising and show a large improvement over the considered baseline.

**Weaknesses:**

However, my main concern is the lack of new scientific conclusions. It seems to me that this work is an iterative improvement over the existing works. From the way I interpret the conclusions and results, authors took FL (already well-studied, including the ASR context) and DP (also relatively well-studied, including in conjunction with FL), merged them together and showed that it can work well on some public datasets.

As large as this increase in performance is (and more on that later), I do not see any methodological novelty here. I would like authors to point me to some concrete steps they undertook compared to prior works in the field, but so far I was not able to find any.

Therefore, while I really enjoyed reading the paper, it is really unclear to me how this work differs from a (very thorough) benchmark on public data using FL+DP?

**Questions:**

It is not clear to me how user-level DP is defined in this work. What is a user with respect to data (e.g. is user - collection of all records with the same ‘author’ or something entirely different?) In fact I could not fully grasp how the number of users go from 50k users to 1M for N?
Additionally, how do you split the data such that it is realistic i.e. so that a user does not have data which is entirely random/disjoint?

Cross-device FL: is it sync/async? What are the implications?

Data splitting: is there an overlap for LS-960? Since the datasets are combined, how do you ensure that the clients are not accounted from twice from privacy budget perspective
How do you ensure a representative data distribution across train-val-test? It was not covered in the dataset description, I am uncertain if your client distribution at test is necessarily similar to the one at test time

Comment on norm layers: these are indeed harder in FL, but I did not find any discussion on this (maybe worth adding how the choice of norm layers affects FL performance in the appendix?)

Seed models improve results: this is expected, the domain shift result is, however, interesting, but also not unusual (if by seed I have correctly interpreted pre-trained models, often on different datasets of the same modality).

The choice of delta-epsilon seems arbitrary: in general i would expect delta to be 1/dataset size, which is not the case for any of the settings, how does this affect accounting?
How exactly is the final epsilon computed (especially considering the per-layer clipping)?

Increasing epsilon is impractical: I would argue that epsilon of 7 is already barely practical given that DP is a multiplicative guarantee.
difficulty of training reasonable models with DP because the impact of noise scales with the model size - I would like the authors to clarify this statement: to me it seems that there is a correlation between some larger models being more prone to performance degradation under DP, but this is correlated with their design rather than their size (e.g. inception net vs resnet)?


Overall, I think there is a lot of work in this manuscript, but I would like the authors to clarify my concerns regarding the scientific contributions as well as some questions on the data distribution/user definition. For now I am really on the fence about recommending acceptance.

---

> ### Author Response · Authors · 2023-11-17
> **Authors' response**
>
> We thank Reviewer JcFR for recognition the quality of our paper and comprehensive experiments we did, their time and feedback. Please find below our comments:
>
> > It is not clear to me how user-level DP is defined in this work. What is a user with respect to data (e.g. is user - collection of all records with the same ‘author’ or something entirely different?) In fact I could not fully grasp how the number of users go from 50k users to 1M for N? Additionally, how do you split the data such that it is realistic i.e. so that a user does not have data which is entirely random/disjoint?
>
> Yes, we consider all audio recordings from the same speaker as one user. For both LS and CV, speaker id is an additional meta information provided in the datasets which we use to group the audio data per speaker (=user).
>
> With user-level DP, the DP guarantee defines the bounds on how much the model can change if we took any user (all data of this user) out of the dataset. To achieve meaningful privacy guarantees when training large models, it is important to work with a large enough population of users. Since CV and LS datasets are not large enough (up to 40K users), we theoretically extrapolated the privacy bounds to assess how much the privacy guarantees would change if the dataset had e.g. 1M users similar to McMahan et al., 2018.
>
> > Cross-device FL: is it sync/async? What are the implications?
>
> We use cross-device FL as described in Algorithm 1 (thus, sync). In each iteration, we randomly sample a cohort of users of specified size, and then wait for all users to finish the training and send model updates, which are then aggregated at the server to update the central model. In a live deployment, each device would typically decide to participate with a given probability and one would typically have to wait for a sufficient number of clients to share model updates until the aggregation can start. Async FL introduces additional level of complexity, is not naturally compatible with differential privacy in the federated setting, and may be considered in the future work.
>
> > Data splitting: is there an overlap for LS-960? Since the datasets are combined, how do you ensure that the clients are not accounted from twice from privacy budget perspective How do you ensure a representative data distribution across train-val-test? It was not covered in the dataset description, I am uncertain if your client distribution at test is necessarily similar to the one at test time
>
> Thanks for the great question! Every split of LS and CV has a separate set of speakers, and validation and test sets also have entirely different speakers from the train. E.g. train-clean-100 and train-clean-360 have a disjoint set of speakers. For LS, dev/test set has 5h with mean of ~8min and std 0.1min for the total duration per user. For CV, dev/test set has ~30h with mean ~15s and std 1.5s for the total duration per user. Thus validation and test data has homogeneous distribution which weights users equally for evaluation. For both datasets we use original validation and test sets, without any modification. Thus, the disjoint set of speakers in different splits and disjoint set of speakers in seed model and FL training ensure that clients are not accounted for twice in privacy budget.
>
> > Comment on norm layers: these are indeed harder in FL, but I did not find any discussion on this (maybe worth adding how the choice of norm layers affects FL performance in the appendix?)
>
> Yes, we have some discussion in Appendix C.2 and some results in Table 7. Let us know if this clarifies the main text and discussion on complexity of post-LN training.
>
> > Seed models improve results: this is expected, the domain shift result is, however, interesting, but also not unusual (if by seed I have correctly interpreted pre-trained models, often on different datasets of the same modality).
>
> Yes, that is correct. What was surprising to us is that domain shift didn’t have much effect on the final model performance.
>
> In contrast, Hsu, et.al. Robust wav2vec 2.0: Analyzing domain shift in self-supervised pre-training, Interspeech 2021 showed that domain mismatch affects the final results for SSL pretraining and Chan, et.al. SpeechStew: Simply Mix All Available Speech Recognition Data to Train One Large Neural Network showed that pretraining on diverse data is helpful, but does not resolve domain shift entirely in ASR.
>
> Another difference with prior works is that we have smaller (# of hours) pre-training dataset that is enough to use for the seed model training and does not affect the final WER. In practice, central data in FL are typically manually curated (possibly public) datasets of smaller size and of different distribution than FL data.

---

> > ### Author Response · Authors · 2023-11-17
> > **Authors' response [continue]**
> >
> > > The choice of delta-epsilon seems arbitrary: in general i would expect delta to be 1/dataset size, which is not the case for any of the settings, how does this affect accounting? How exactly is the final epsilon computed (especially considering the per-layer clipping)?
> >
> > The smaller the delta, the worse the epsilon, so the two parameters are related. It is recommended that delta is much smaller than 1/population size. Following McMahan et al., 2018, we thus decided to fix one of them (delta) to $10^{-9}$ (much smaller than 1/population size regardless of the scaling). Final epsilon is computed with Gaussian moments accountant method using opacus with the delta set to $10^{-9}$.
> >
> > Per-layer clipping guarantees the total bound on client model updates to be clipping bound C, thus it mathematically doesn’t matter whether we do per-layer clipping or global clipping (we explain this in the last paragraph of Section 3).
> >
> > > Increasing epsilon is impractical: I would argue that epsilon of 7 is already barely practical given that DP is a multiplicative guarantee. difficulty of training reasonable models with DP because the impact of noise scales with the model size - I would like the authors to clarify this statement: to me it seems that there is a correlation between some larger models being more prone to performance degradation under DP, but this is correlated with their design rather than their size (e.g. inception net vs resnet)?
> >
> > * Without extrapolation by scaling the population size and cohort size, the large epsilons would indeed be impractical. We think that the values of epsilon can be further decreased in the future; our baseline can serve as a good starting point.
> > * As the number of parameters in a model increases, clipping has an increased impact on scaling down each parameter (factor of approximately 1/sqrt(dim)). With constant noise per parameter, the signal to noise ratio thus decreases with the number of model parameters and consequently the optimization becomes more complex.
> > * Interestingly, our work used larger models (on purpose) than in prior work on FL for ASR and it achieved better results nonetheless. However, the large number of parameters did not make it infeasible to introduce DP.
> >
> > > However, my main concern is the lack of new scientific conclusions. It seems to me that this work is an iterative improvement over the existing works. From the way I interpret the conclusions and results, authors took FL (already well-studied, including the ASR context) and DP (also relatively well-studied, including in conjunction with FL), merged them together and showed that it can work well on some public datasets.
> >
> > > As large as this increase in performance is (and more on that later), I do not see any methodological novelty here. I would like authors to point me to some concrete steps they undertook compared to prior works in the field, but so far I was not able to find any.
> >
> > > Therefore, while I really enjoyed reading the paper, it is really unclear to me how this work differs from a (very thorough) benchmark on public data using FL+DP?
> >
> > We refer to the general response to all reviewers.

---

> > > ### Comment · Reviewer_JcFR · 2023-11-22
> > > **Response to authors**
> > >
> > > I would like to thank the authors for their comprehensive response to my queries!
> > >
> > > However, I still firmly believe that despite the impressive results and the fact that, this work may be great from engineering perspective, there is hardly anything novel from the scientific perspective unfortunately. So my score remains unchanged

---

### Official Review · Reviewer_4uMy · 2023-10-31

**Soundness:** 2 fair
**Presentation:** 3 good
**Contribution:** 2 fair
**Rating:** 3
**Confidence:** 4

**Summary:**

In this paper, the authors focus on (i) providing an empirical analysis of FL performance on E2E ASR using a large model trained with the Connectionist Temporal Classification (CTC) loss; (ii) analyzing the impact of key FL parameters, data heterogeneity, and seed models on FL performance in ASR; (iii) formulating a practical benchmark and establishing the first baselines for FL with DP for ASR; and (iv) reviving the per-layer clipping for efficient training of large transformer models even in the presence of DP noise.

**Strengths:**

The quality of this paper is good, but the originality, clarity and significance indeed need to be improved.
1.    Originality. This paper proposes to apply DP-federated learning to the ASR task, which has not been done before. However, DP for federated learning has already been done long ago, and this work only tests the algorithm on the data instead of improving the frame. Maybe there is a small originality but it’s not enough for ICLR.
2.     Quality. This article conducted a substantial number of experiments to investigate the impact of seed models, data heterogeneity, and differential privacy on the performance of federated learning. And the content and structure of the paper are relatively complete.
3.	Clarify. The content and structure of the article are relatively comprehensive. However, the experiment logic may not be very clear. For example, from the algorithm pseudocode, it appears that this article only combines DP (Differential Privacy) and FedAvg (Federated Averaging). Yet, later in the text, there is an experimental analysis of Scaffold and FedProx, showing that FedProx yields the best results. So, what is the significance of exploring data heterogeneity? The article could benefit from providing a more explicit and coherent explanation of the rationale for the experimental choices and their implications for understanding data heterogeneity.
4.	Significance. Well this work is the first to combine federated learning with differential privacy in the ASR (Automatic Speech Recognition) field, which can be considered pioneering. The choice of this application direction is valid, but it appears that the level of innovation is relatively limited, which might diminish its overall importance.

**Weaknesses:**

The level of innovation in this paper is insufficient, not absent but lacking. The combination of federated learning and differential privacy is something that has been explored before, as mentioned in your paper. I suggest further research to discover if there is a federated learning algorithm specifically suited for ASR. ICLR is a highly prestigious conference, and you need to enhance the innovation of your paper in terms of algorithms.
The paper lacks innovation in terms of algorithms.
The logical relationships between experiments are not clear, and there is a weak connection between the models involved in the experiments and the models proposed in paper.
The epsilon in the experiment is too large, around 10^8.

**Questions:**

It seems that all experimental results in this paper have not been averaged over multiple runs, which introduces some randomness.
Table 2 has many ‘-‘. Why not delete the lines that are intractable?

---

> ### Author Response · Authors · 2023-11-17
> **Authors' response**
>
> We thank Reviewer 4uMy for recognition the quality of our work, their time and feedback. Please find below our comments.
>
> > Originality. This paper proposes to apply DP-federated learning to the ASR task, which has not been done before. However, DP for federated learning has already been done long ago, and this work only tests the algorithm on the data instead of improving the frame. Maybe there is a small originality but it’s not enough for ICLR.
>
> We refer to the general response to all reviewers.
>
> > Clarify. The content and structure of the article are relatively comprehensive. However, the experiment logic may not be very clear. For example, from the algorithm pseudocode, it appears that this article only combines DP (Differential Privacy) and FedAvg (Federated Averaging). Yet, later in the text, there is an experimental analysis of Scaffold and FedProx, showing that FedProx yields the best results. So, what is the significance of exploring data heterogeneity? The article could benefit from providing a more explicit and coherent explanation of the rationale for the experimental choices and their implications for understanding data heterogeneity.
>
> Thanks for the feedback! We list only final **simplest** version in Algorithm 1, which throughout the paper we demonstrate is enough to tackle the data heterogeneity for ASR task with proper combination of hyper-parameters and adaptive optimizer. We discuss other techniques, like scaffold and fedprox, as ablations and show consistent results that for **large** ASR models adaptive optimizers resolve data heterogeneity issue as opposite to prior works. See details e.g. in Appendix C.3 and C.6.
>
> > Significance. Well this work is the first to combine federated learning with differential privacy in the ASR (Automatic Speech Recognition) field, which can be considered pioneering. The choice of this application direction is valid, but it appears that the level of innovation is relatively limited, which might diminish its overall importance.
>
> Our main contributions are:
> * via empirical analysis we show inconsistent results with respect to prior works on FL, FL+DP, and FL specifically for ASR, e.g. (1) data heterogeneity does not prevent FL from finding competitive models when using adaptive optimizers even for simple FedAvg; (2) FL for ASR can train competitive ASR models even (a) from scratch **unlike prior works** or (b) using a seed model trained on **out-of-domain** data; and (3) per-layer clipping provides a significant performance boost for FL+DP in ASR.
> * to the best of our knowledge, we are the first to successfully train FL / FL + DP ASR model with simplest known algorithm (FedAvg + gaussian moments accountant) and practical settings (2k central steps, realistic data distribution, out-of-domain central data).
>
> We truly think that our empirical observations and new benchmark has their place at ICLR to contribute to community knowledge and foster development of general FL and FL + DP algorithms generalizable across domains including ASR.
>
> > The level of innovation in this paper is insufficient, not absent but lacking. The combination of federated learning and differential privacy is something that has been explored before, as mentioned in your paper. I suggest further research to discover if there is a federated learning algorithm specifically suited for ASR.
>
> > ICLR is a highly prestigious conference, and you need to enhance the innovation of your paper in terms of algorithms. The paper lacks innovation in terms of algorithms.
>
> Given the prior works on FL for ASR, it would indeed seem to be necessary to develop new FL algorithms to successfully tackle ASR with FL (see Section 2). However, we show empirically that this is **unnecessary** and with the **right combination of existing simple (e.g. FedAvg) components which we analyze in depth**, FL and FL+DP is capable of finding competitive ASR models in many scenarios.
>
> > The logical relationships between experiments are not clear, and there is a weak connection between the models involved in the experiments and the models proposed in paper.
>
> Could Reviewer clarify this statement and be concrete about particular experiments and models?

---

> > ### Author Response · Authors · 2023-11-17
> > **Authors' response [continue]**
> >
> > > The epsilon in the experiment is too large, around 10^8.
> >
> > It is well known that DP needs large batches and large population to achieve practical DP guarantees. The main obstacle in ASR is to scale to e.g. 7M speakers (which is a practical number for mid-resource languages) as we are not aware of any publicly available ASR dataset of that scale. For that reason we theoretically extrapolate DP guarantees assuming the training dynamic remains unchanged with scale of 7M population and get practical epsilon < 10, similar to prior works e.g. McMahan et al., 2018, McSherry, F. and Mironov, I., 2009, June. Differentially private recommender systems: Building privacy into the netflix prize contenders. ACM SIGKDD. Besides, we set an initial benchmark for community showing its challenges and training dynamic to foster development of FL + DP in ASR.
> >
> > > It seems that all experimental results in this paper have not been averaged over multiple runs, which introduces some randomness. Table 2 has many ‘-‘. Why not delete the lines that are intractable?
> >
> > Each experiment in ASR is expensive to run even for central training. Instead, we run many ablations and training for different combinations of seed models and data. We also confirmed hyper-parameters robustness by applying **exactly the same parameters** from English data to French and German data.
> >
> > The ‘-’ symbols in Table 2 are extrapolations (similar to McMahan et al., 2018, McSherry, F. and Mironov, I., 2009, June. Differentially private recommender systems: Building privacy into the netflix prize contenders. ACM SIGKDD) that define the change in privacy guarantees obtained for larger population and cohort sizes, assuming that scaling laws are in place giving similar WER with scaling both population and cohort size by 100x.

---

> > > ### Comment · Reviewer_4uMy · 2023-11-23
> > > **Thanks for the response**
> > >
> > > I'd like to thank authors for detailed response. Unfortunately, considering for the novelty and contribution, I keep my scores. I hope to see it be an improved version in the future.

---

### Author Response · Authors · 2023-11-17
**General response to all Reviewers**

Dear Reviewers,

We would to clarify general questions raised by several of you at the same time:

* While we agree with the feedback that our work essentially combines several known algorithms (pre-LN transformer, LAMB optimizer, FedAvg, DP, ASR) it is non-trivial as evident from prior works that fail to show convergence with such a setup. Our claim to novelty does not lie in the development of new FL, DP, or ASR algorithms, but in achieving a **non-trivial setup that is competitive with centralized training.**
* We would like to highlight that the domain of FL for ASR is a particularly challenging one as evident from (1) the prior works on FL for ASR (which show that FL for ASR does not seem feasible in a practical setting) and (2) the fact that FL algorithms have been deployed for various use cases including vision and NLP but none (we are aware of) for ASR specifically yet.
* Instead of algorithmic contribution we bring novelty on the empirical side, practical side and new SOTA results for ASR in FL and FL+DP:
    * Via empirical analysis we highlight inconsistencies with respect to results from prior works (discussed in Section 2) on FL, FL+DP, and FL specifically for ASR, e.g. (1) data heterogeneity does not prevent FL from finding competitive models when using adaptive optimizers even for simple FedAvg; (2) FL for ASR can train competitive ASR models even (a) from scratch **unlike prior works** or (b) using a seed model trained on **out-of-domain data**; and (3) per-layer clipping provides a significant performance boost for FL+DP in ASR.
    * No prior work on FL in ASR demonstrated competitive performance in scenarios that could enable practical FL deployment; either the performance was too poor and required seed models from the same dataset, or the number of central step was not practical for FL. To the best of our knowledge, we are the first to successfully train FL / FL + DP ASR model with **practical settings** (2k central steps, realistic data distribution, out-of-domain central data) using the **simplest known algorithm** (FedAvg + gaussian moments accountant).
    * We showed that the proposed combination of methods works out of the box on data from other languages without any additional tuning, providing support for the robustness of our empirical findings.
    * We show empirically that this it is **unnecessary** in the first place to develop new FL algorithms for ASR, and **the right combination of existing simple (e.g. FedAvg) components which we analyze in depth** is capable of finding competitive ASR models in many scenarios.
    * We revived per-layer clipping and provided deep empirical analysis (Appendix D.5 and D.6) on when and why it is necessary for FL + DP in ASR in contrast to prior works which observe only marginal influence.
    * Lastly, to the best of our knowledge, we are the first to provide deep empirical analysis for the **large-scale models** in FL trained from scratch as well as first benchmark for FL + DP in ASR.
* We believe that **establishing a competitive baseline** often forms the backbone for advancing research in a field and that our paper through the exposure of a venue like ICLR would help put the spotlight on ASR for FL/DP which is an overlooked benchmark by most existing works in FL/DP.